# INTERPOLATE: HOW RESETTING NEURONS WITH MODEL INTERPOLATION CAN IMPROVE GENERALIZABILITY IN ONLINE LEARNING

## ABSTRACT

While neural networks have shown a significant gain in performance across a wide range of applications, they still struggle in non-stationary settings as they tend to lose their ability to adapt to new tasks — a phenomenon known as the loss of plasticity. The conventional approach to addressing this problem often involves resetting the most under-utilized or dormant parts of the network, suggesting that recycling such parameters is crucial for maintaining a model's plasticity. In this study, we explore whether this approach is the only way to address plasticity loss. We introduce a resetting approach based on model merging called Interpolate and show that contrary to previous findings, resetting even the most active parameters using our approach can also lead to better generalization. We further show that Interpolate can perform similarly or better compared to traditional resetting methods, offering a new perspective on training dynamics in non-stationary settings.

## 1 INTRODUCTION

Recent advancements in deep learning have significantly improved the performance of neural networks across a wide range of tasks (Miikkulainen et al., 2024). However, as the volume of training data continues to grow, the importance of online learning becomes increasingly evident (Dohare et al., 2024). Unlike traditional training methods that rely on independent and identically distributed (i.i.d.) data, online learning allows models to continuously adapt to new information, making them more robust to the ever-changing nature of the real-world (Lyle et al., 2023; Elsayed & Mahmood, 2024). However, training neural networks in non-i.i.d. settings introduces new challenges, such as catastrophic forgetting, where the model tends to forget past information (Goodfellow et al., 2013; Kim & Han, 2023) and loss of plasticity, where the model's ability to learn new tasks decreases (Ash & Adams, 2020; Kim et al., 2023).

Numerous methods have been proposed in the literature to address plasticity loss, such as resetting parameters based on the neuron's activity (Dohare et al., 2021), regularizing based on parameter norm and gradient norm (Kumar et al., 2023; Lewandowski et al., 2024a), and modifying the model architecture (Abbas et al., 2023). However, Lyle et al. (2024) recently showed that no single method is sufficient to fully mitigate the loss of plasticity.

Among these methods, dormancy in neurons is often correlated with loss of plasticity, but it is not the direct cause (Lewandowski et al., 2024b). However, existing plasticity methods in deep neural networks mainly rely on resetting the dormant parameters of the selected network using criteria such as dormancy scores (Sokar et al., 2023). The intuition behind this approach is to recycle dormant neurons back into an *active* state to recover some of the network's capacity. However, research has shown that dormancy does not always correlate with a loss of plasticity. While resetting dormant neurons helps in trainability, it is still outperformed in terms of generalizability by methods like shrink and perturb (S&P) (Ash & Adams, 2020), which involves adding noise to the parameters. It raises the question: Can resetting the non-dormant neurons also improve plasticity? How many parameters should be reset? Moreover, is resetting the parameters associated with dormant neurons the only method to *reactivate* the model? Exploring alternative strategies could lead to more effective ways to understand deep neural network dynamics in online learning.

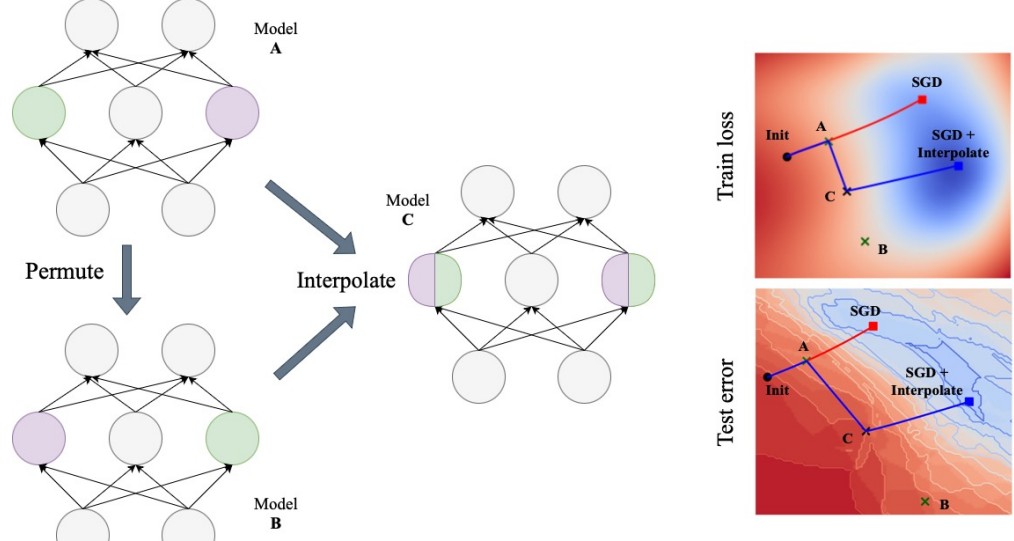

Figure 1: Our proposed model-merging approach *Interpolate* for resetting model parameters in non-stationary settings. We utilize the permutation invariance property in neural networks (Entezari et al., 2021) and merge a given base model A with its unique functionally equivalent permuted variant model B in which green and purple hidden nodes were selected to be permuted. Next, we obtain model C which is combination of A and B (linear interpolation) and train the model. The 2D contour plots of Train loss and Test error surfaces illustrate the resulting trajectory for training from A and C (Li et al., 2018). Training from C (blue) resulted in discovery of a generalizable region in the loss surface as compared to training from A (red).

We investigate existing plasticity methods that use different utility functions to select neurons for reset. Note that, by *reset*, we specifically mean re-locating the parameters on a different point in the loss landscape. Therefore, throughout our paper, the term *reset* encompasses any type of modification on model parameters and is not limited to re-randomization or re-initialization.

With the goal of improving generalizability rather than only trainability, we explore model merging as an alternative way to reset the model parameters (Wortsman et al., 2022; Yang et al., 2024). Our motivation comes from an extensive literature on linear mode connectivity and loss barrier analysis which suggests a link between low-loss barriers between minima with training stability and generalization (Frankle & Carbin, 2018). Several approaches have been proposed to improve linear mode connectivity by reducing loss barriers between minima in order to improve generalization through model merging techniques (Mirzadeh et al., 2020; Tatro et al., 2020). However, Entezari et al. (2021) showed that such loss barriers between minima can be minimized cost-effectively by exploiting the permutation invariance property of neural networks. By resetting the model on high-barrier regions, we propose our method Interpolate which utilizes permutation invariance to reset highly active parameters in non-stationary settings which intentionally introduces controlled instability, acting as a regularizer. We hypothesize that training from this reset point would allow SGD to navigate toward a more stable loss region, ultimately improving generalization. Figure 1 summarizes our overall idea on how model-merging with permutation invariance property can help in finding generalizable regions in the loss surface which essentially challenges the prevailing narrative in the plasticity research community that predominantly focus dormant neurons.

We summarize our contribution as follows:

- In contrast to previous findings that plasticity requires resetting inactive parameters, our analysis reveals that resetting the most active parameters can yield similar improvements.

- We introduce a model-merging method called Interpolate, leveraging the permutation invariance property in neural networks to offer a new perspective on the resetting techniques used for addressing plasticity loss.

- We provide empirical results using Interpolate across various distribution shifts on MLP and CNN models, demonstrating that it can achieve performance comparable to, or even better than, existing baselines.

The rest of the paper is organized as follows. We discuss the related work in section 2 and provide a brief background on plasticity and model merging. We describe our proposed method to reset model parameters for maintaining plasticity in section 3. This is followed by the experiments in section 4 where we provide all our results along with analysis and conclusions in section 5.

## 2 BACKGROUND

### 2.1 PLASTICITY

Plasticity refers to a neural network's ability to adapt to new tasks when the data distribution shifts. Several metrics have been proposed to quantify the loss of plasticity (Dohare et al., 2021; Lyle et al., 2023; Lee et al., 2024). Following Lee et al. (2024), we measure the loss of plasticity using the test accuracy of the model on the final task in online learning setups.

Numerous studies have explored potential causes for the loss of plasticity in deep learning models when used in non-stationary settings. Existing approaches to mitigating this issue can be classified into three categories: (i) reset-based methods, (ii) regularization-based methods, and (iii) architecture-based methods. These categories are orthogonal to each other and thus can be combined to achieve superior performance. While our focus is on reset-based methods as we study resetting active parameters, we briefly outline all three categories in this section to provide an overview of existing approaches.

**Reset-based methods** This class of methods involves selectively resetting a subset of model parameters with the goal of reviving the model's plasticity (Igl et al., 2020; Nikishin et al., 2022). They usually comprise two key elements: a utility function and a reset function. While several types of utility and reset functions have been explored in the literature, a common assumption is that randomly reinitializing inactive neurons is essential for restoring plasticity. Two of the most popular methods that follow this assumption are Recycling Dormant Neurons (ReDo) (Sokar et al., 2023), which uses activation scores as its utility function, and Continual Backprop (CBP) (Dohare et al., 2021), which uses a maturity threshold as its utility function. These methods are discussed in detail later in section 3. In this work, we analyze different utility functions and propose a reset function that demonstrates how resetting to *active* neurons of the model can also help prevent plasticity loss.

**Regularization-based methods** These methods control the training dynamics in online learning by regulating factors such as weight norm, gradient norm, or spectral norm (Lewandowski et al., 2024a). Lyle et al. (2023) conducted an empirical analysis revealing that plasticity loss is closely related to changes in the curvature of the loss landscape. Lewandowski et al. (2024b) also introduced a regularization method to preserve curvature across different dimensions to mitigate plasticity loss. Alternatively, Kumar et al. (2023) proposed a regularization approach similar to L2 but penalizing with respect to the initial parameters called L2 Init.

**Architecture-based methods** Another class of methods focuses on modifying model components to overcome problems that cause plasticity loss. Abbas et al. (2023) associated the plasticity loss problem with an increase in the number of dead neurons due to the presence of ReLU activation functions and proposed an alternate activation function called CReLU to prevent activation collapse. Lyle et al. (2024) suggested that using layer normalization (Ba et al., 2016) with L2 regularization to maintain low activation and weight norms improves generalization performance across several benchmarks.

**Other plasticity methods** Lyle et al. (2024) also investigated how different mechanisms of plasticity loss can be effectively combined and demonstrated that addressing multiple mechanisms simultaneously, rather than focusing on a single one, leads to highly robust learning algorithms. One example of such a method is Utility-based Perturbed Gradient Descent (UPGD) (Elsayed & Mahmood, 2024), which applies smaller gradient updates to more useful units to preserve past

knowledge while applying larger updates to less useful units to increase their plasticity. Ash & Adams (2020) proposed Shrink & Perturb where all parameters are updated by decaying weight magnitude and adding small random noise to them. This approach is also known to improve generalizability better compared to other methods apart from trainability. Lee et al. (2024) explored warm-starting experiments from Ash & Adams (2020) further and introduced the Hare & Tortoise approach that involves periodically replacing the fast weights with the slow weights.

While these methods improve both trainability and generalizability aspects of the model under non-stationary settings, they are orthogonal to our analysis of utility and reset functions.

## 2.2 MODEL MERGING

Generalization performance in neural networks is significantly influenced by how optimizers navigate the loss landscape. Sun (2019) suggested that these landscapes may possess simple, non-trivial properties that can be leveraged to improve performance. One such property that recently gained interest in the machine learning community is linear mode connectivity which involves linearly interpolating two independently trained models (Lee & Lee, 2024; Vlaar & Frankle, 2022).

Several studies have demonstrated merging pre-trained models in this manner can result in a model with greater generalization capabilities (Wortsman et al., 2022; Zhou et al., 2023). Moreover, Yang et al. (2024) also showed that this approach can be utilized for efficient knowledge transfer between existing large language models without training them on additional data.

Another important property of neural networks that has been explored in the context of model merging and mode connectivity is permutation invariance (Ganju et al., 2018; Entezari et al., 2021; Simsek et al., 2021). This property states that fully connected neural networks are invariant to the permutation of neurons within hidden layers. In other words, permuting the weights associated with these neurons yields a functionally equivalent network. Ainsworth et al. (2023) leveraged this property and introduced multiple algorithms to permute neurons of a given model to align them with a reference model with the goal of merging them in weight space.

In our work, we argue that, under non-stationary settings, model merging using permutation invariance can serve as an effective resetting function. Unlike traditional resetting methods that often discard older knowledge and require relearning from random noise, we argue that model merging can exhibit better knowledge transfer for future tasks which is essential for maintaining generalizability over time.

## 3 METHODOLOGY

In this section, we introduce Interpolate, a reset-based method, which consists of two key components: (i) selecting the most active neurons and (ii) a novel reset method that uses model merging. We will describe each of these components in detail.

### 3.1 HOW TO SELECT NEURONS?

Unlike previous reset-based methods, our approach focuses on selecting and resetting active neurons within the model. We employ the *dormancy score* utility function proposed for ReDo (Sokar et al., 2023): let $h_i(x)$ correspond to the activation of the neuron with index $i$ in a layer with $L$ neurons when the network is given input $x$. For a given neuron $i$, its dormancy on dataset $D$ is defined as:

$$d_i = \frac{\mathbb{E}_{x \in D}|h_i(x)|}{\frac{1}{L}\sum_{j=1}^{L}\mathbb{E}_{x \in D}|h_j(x)|} . \tag{1}$$

In ReDo, neuron $i$ is selected to be reset if $d_i \leq \tau$, where $\tau$ is a hyper-parameter called the dormancy threshold.

We also compute the dormancy score for each neuron similar to ReDo. To validate our hypothesis about resetting most active neurons, however, instead of selecting neurons with scores below a certain threshold $\tau$, we choose the neurons based on top $k$ percentile of $d_i$. We denote the parameters corresponding to the selected neurons as $\hat{\theta}_k$.

## 3.2 How to reset?

To reset the neurons selected based on the utility function, both ReDo and CBP re-initialized the neuron's input weights randomly (using the same distribution as the network initialization) and set the neuron's output weights to zero, ensuring that the new model state does not alter the output. Although other techniques exist for resetting parameters in some specific parts of the model (Nikishin et al., 2022), researchers tend to prefer resetting with random noise in online learning to mitigate plasticity loss and often use ReDo and CBP as their baselines (Abbas et al., 2023; Dohare et al., 2024). We propose a novel approach for resetting active neurons motivated by the permutation invariance property in neural networks which has been explored previously in the deep learning literature (Ganju et al., 2018).

Let $\mathbf{P}_{\hat{\theta}_k}$ represent the set of all valid permutations that result in functionally equivalent parameters to network parameters $\theta = (\theta_1, \theta_2, ..., \theta_d)$ by randomly permuting parameters in the subset $\hat{\theta}_k$ among themselves. This allows us to define the permutation function as $P : \mathbb{R}^d \times \mathbf{P}_{\hat{\theta}_k} \rightarrow \mathbb{R}^d$ (Entezari et al., 2021; Simsek et al., 2021). We can thus obtain a new *permuted* parameter configuration $P(\theta, \pi_k) = \theta_{perm} = (\theta_{\pi_k(1)}, \theta_{\pi_k(2)}, ..., \theta_{\pi_k(d)})$ by applying permutation $\pi_k \sim \mathbf{P}_{\hat{\theta}_k}$ to the subset of parameters $\hat{\theta}_k \subseteq \theta$. This $\theta_{\text{perm}}$ is functionally equivalent to $\theta$, i.e. $\mathcal{L}(\theta_{\text{perm}}) = \mathcal{L}$. Finally, to obtain our reset network, we simply merge the models by finding the midpoint between $\theta$ and $\theta_{\text{perm}}$:

$$\theta_{\text{reset}} = \frac{\theta_{\text{perm}} + \theta}{2} \tag{2}$$

This approach can be viewed as merging two equivalent models that share the same functional properties within their local regions in the loss landscape. By combining these models, the parameters are effectively shifted to a region with a higher loss value, as the most active neurons are reset, resulting in the *unlearning* of those parameters. Therefore, when the new batch arrives, these dimensions will be re-learned and as a result, the new gradients with higher magnitudes would perturb other dimensions, potentially improving the overall adaptability and performance of the model. Although this *unlearning* technique may appear counter-intuitive, such behavior was previously observed in the analysis by Vlaar & Frankle (2022), which suggested that initializing a model on a higher loss surface—obtained from the height of the barrier in the linear interpolation of models, rather than using random initialization—led to a network achieving better test accuracy. In our experiments, we will demonstrate that Interpolate acts as an adversarial technique, resulting in performance comparable to or better than conventional ReDo. We provide the pseudo-code in Algorithm 1.

---

**Algorithm 1** Interpolate to reset

---

**Require:** Input dataset $D$, Base model parameters $\theta$, $k$ percentile
  Apply forward pass on model $\theta$ with $D$ and store activation outputs of all neurons in $H$
  $\mathbf{d} \leftarrow \{\}$
  **for** $i = 1, 2, \ldots, |H|$ **do**
    Compute dormancy score $d_i$ using equation 1
    Append $d_i$ in $\mathbf{d}$
  **end for**
  $K \leftarrow$ list indices of top $k$ percentile values in $\mathbf{d}$
  $\hat{\theta}_k \leftarrow \theta[K]$
  Sample $\pi_k$ from $\mathbf{P}_{\hat{\theta}_k}$ without replacement
  $\theta_{\text{perm}} \leftarrow P(\theta, \pi_k)$
  **return** $(\theta_{\text{perm}} + \theta)/2$

---

## 4 Experiments

We provide a series of experiments that reveal how resetting active neurons can achieve comparable performance, showing that recycling inactive neurons is not the only way to restore a model's plasticity. We use three types of distribution shifts on CIFAR10 dataset (Krizhevsky et al., 2009): (i) Shuffled (Lewandowski et al., 2024a), where the labels are randomly flipped for each task; (ii)

Noisy (Lee et al., 2024), where each task is a subset dataset and contains decreasing levels of label noise; (iii) Permuted (Goodfellow et al., 2013), where the input data is randomly permuted for each task.

We start with an empirical analysis to compare utility and reset functions, including Interpolate, by fixing the number of neurons in subsection 4.1 and subsection 4.2 on the Shuffled CIFAR10 benchmark. We also provide a brief sensitivity analysis to demonstrate the benefits of combining Interpolate with ReDo in subsection 4.3. Finally, in subsection 4.4, we conduct an extensive hyper-parameter search and show that interpolating active neurons can match the performance of several state-of-the-art baselines. We use an MLP with 3 hidden layers, each consisting of 128 neurons. We also use CNN for the hyper-parameter search experiment which consists of 2 convolutional layers with 16 filters. All experimental results involve five random seeds.

## 4.1 INTERPOLATE VS RANDOM NOISE

We study how reset by Interpolate helps bring the model into an active state to mitigate the loss of plasticity. We compare it with reset by random noise obtained using Lecun normal initializer (Bradbury et al., 2018). We train the MLP on Shuffled CIFAR10 with up to 50 tasks and 500 epochs per task. Next, we train this model on a new task for 100 epochs, at which point we randomly select a given number of neurons, apply reset, and then train this updated model until convergence.

In Figure 2, we compare the best generalization performance obtained when increasing the number of selected neurons for both strategies. On average, the performance of Interpolate is better than random noise. Additionally, there is a slightly positive correlation between the number of interpolated neurons and performance, suggesting that as more neurons are interpolated, the model adapts more seamlessly to the new task without compromising prior knowledge. In contrast, random noise shows a negative correlation with performance, as increasing the number of randomly initialized neurons introduces instability leading to relatively worse performance. Figure 2 (right) shows the jump in training loss which is the difference between training loss computed just before and after resetting the parameters using Interpolate or random noise. Random noise results in a higher jump in loss as more neurons are affected, indicating greater forgetting, whereas interpolation has a less detrimental impact on the model's internal representations.

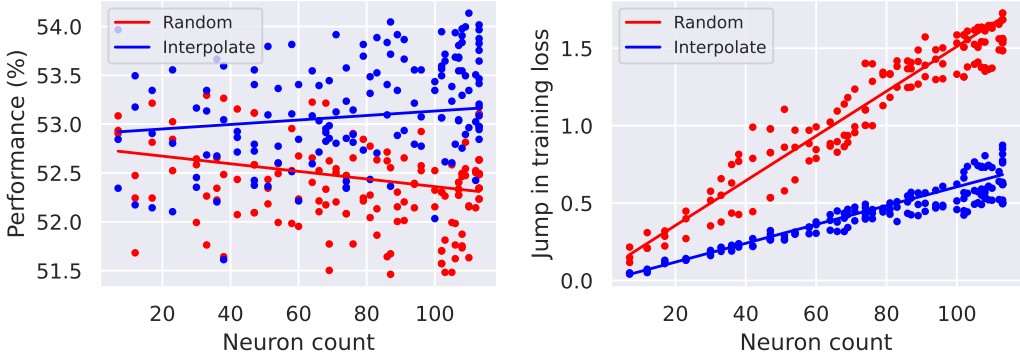

Figure 2: Comparing generalization performance (left) and jump in training loss (right) for random noise and Interpolate reset functions. Interpolate results in relatively more efficient adaptation to new tasks, while random noise can introduce instability and performance loss when applied to too many neurons.

## 4.2 RESETTING ACTIVE VS INACTIVE NEURONS

Next, we investigate whether selecting the most active neurons for resetting can also improve generalization in an online learning setup. We compare ReDo and Interpolate, using top $k$ percentile and bottom $k$ percentile dormancy scores as the utility functions. The goal is to understand how these methods affect online test accuracy, dormancy, weight norm, and gradient norm over multiple tasks. We evaluate the methods on Shuffled CIFAR10 tasks where each task is trained for 10 epochs. For

ReDo and Interpolate, the labels in Figure 3 indicate the $k\%$ of total neurons selected for resetting, based on their dormancy score – top-$k$ (active) or bottom-$k$ (inactive), where $k \in \{5\%, 20\%\}$.[1] The reset period is fixed at 5 epochs on each experiment. We also plot results obtained using CBP and Interpolate (CBP) where we use CBP's utility function and apply Interpolate to reset instead of random noise.

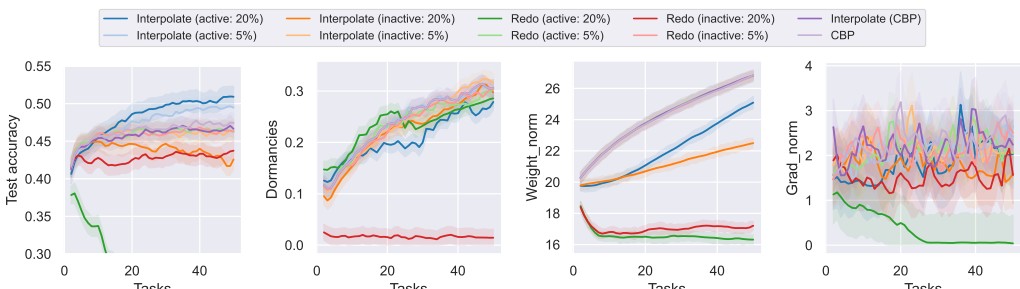

Figure 3: Comparing ReDo and interpolation performs with active/inactive neurons without any hyper-parameter search on Shuffled CIFAR10 with MLP. Applying Interpolate on active neurons results in the highest performance gain even when the dormancy is higher. On the other hand, ReDo results in relatively worse performance even with lower dormancy and lower weight norm. Resetting more active neurons has a catastrophic effect on the learning process as the gradient norm diminishes.

There are several interesting trends observed. While interpolating the inactive neurons i.e., both Interpolate (inactive) and Interpolate (CBP), do not result in the best overall performance, Interpolate (active) improves over ReDo (active), ReDo (inactive) and CBP. This contradicts the common intuition that only resetting inactive neurons would help in utilizing the model's capacity. In fact, these results suggest that resetting active neurons can also lead to a competitive performance.

ReDo (inactive) also results in lower dormancy (in Figure 3 (second)), but this does not correlate with higher performance. On the other hand, Interpolate (active) does not decrease dormancy but still results in better performance, which again challenges the idea of reviving dormant neurons. We further observe that ReDo, which resets neurons by setting output weights to zero, results in a lower weight norm, unlike interpolation, which does not control the weight norm significantly but still outperforms. However, the increasing weight norm problem in non-stationary settings has already been addressed with L2 regularization (Ash & Adams, 2020; Dohare et al., 2021). In terms of gradient norm, ReDo (active) leads to a significant drop as observed in Figure 3 (forth), indicating that no meaningful learning occurs, which is detrimental to the model's performance. Furthermore, we observe that the gradients obtained by using Interpolate (active: 20%) have higher magnitude as compared to say Redo (inactive: 20%) which resets most dormant neurons. This validates our hypothesis that the resulting gradients in Interpolate perturbs all dimensions, potentially improving the overall adaptability and performance of the model. Overall, we also conclude that while no single metric can fully explain the performance trends, resetting inactive neurons is not the only way to revive the model's plasticity.

### 4.3 COMBINING INTERPOLATE (ACTIVE) WITH REDO (INACTIVE)

Since both strategies, ReDo (inactive) and Interpolate (active) work well individually, we now explore how their combination would perform for a fixed number of neurons. Specifically, we investigate how different ratios of neurons reset using these strategies impact model performance.

This analysis uses the same setup as the previous one, with default hyper-parameters but different compute budgets. We also add the Noisy CIFAR10 dataset for our analysis. In each scenario, we vary the percentile of neurons ($k$) selected for ReDo and apply Interpolate to the remaining neurons. The reset period is fixed at 5 epochs on each experiment, and we compare performance for increasing $k$.

Figure 4 shows the online test accuracy observed. When training for 10 epochs per task, applying Interpolate consistently improves performance compared to ReDo. This indicates that for both Noisy

---

[1]These values of $k$ were chosen because they are commonly used in the literature as default. These values have also shown competitive performance in our experiments discussed later.

and Shuffled CIFAR10, the model benefits more from interpolating neurons rather than resetting them.

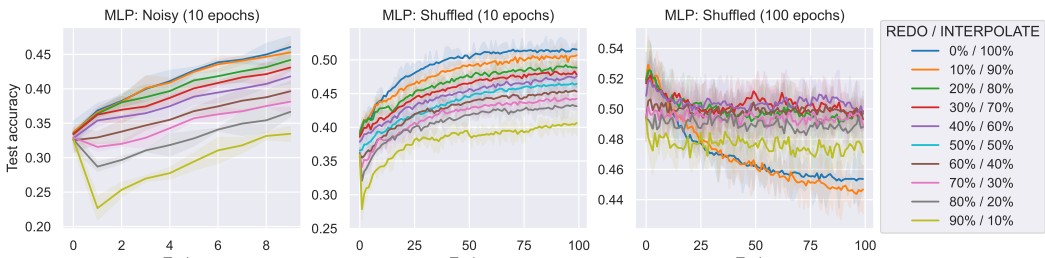

Figure 4: Comparing online test accuracy for resetting $k\%$ least active neurons with ReDo and simultaneously applying Interpolate on the remaining neurons on Shuffled and Noisy CIFAR10 with MLP. We observe that as $k$ increases, the performance degrades indicating a clear advantage of using Interpolate over ReDo for less compute budget. For a higher compute budget (100 epochs per task), there is an optimal balance between Interpolate and ReDo where $k$ lies between 30 to $40\%$.

For 100 epochs per task on Shuffled CIFAR10, while performance generally improves as more neurons are interpolated rather than reset, the model underperforms when nearly all neurons are interpolated. This suggests that there is an optimal balance between ReDo and Interpolate. The best performance occurs when $30 - 40\%$ of neurons are reset and $60 - 70\%$ are interpolated. In all scenarios, ReDo with $90\%$ of neurons results in poor performance, which is expected since excessive resetting would hurt the model's ability to retain useful learned knowledge. Overall, while Interpolate improves performance, finding the right balance between the number of neurons for ReDo and Interpolate is crucial for optimal results when these methods are combined, especially in larger epoch settings. While in these experiments reset the whole model, we also conducted experiments with an exhaustive hyper-parameter search for varying combination of number of neurons selected for ReDo and Interpolate in Appendix A.3.9.

## 4.4 COMPARING WITH BASELINES

The previous analysis indicated how Interpolate (active) can potentially achieve comparable performance as ReDo which involves resetting the under-utilized and inactive parts of the model. In this experiment, we investigate whether Interpolate can still result in a similar performance as other plasticity baselines after an exhaustive hyper-parameter search is applied for model selection.

The experiments are conducted on Shuffled, Permuted, and Noisy CIFAR10 settings, using MLP and CNN architectures. The models are optimized using SGD with L2 regularization. We compare Interpolate and Interpolate+ReDo with the following plasticity baselines: CBP, ReDo and naive SGD. We also use Reinit (Full) as an additional baselines where we re-initialize the whole model at the beginning of each task. The optimal hyper-parameter setup is selected through a random search over all possible configurations. For each method, the search is limited to a maximum of 20 configurations, with the best setup selected based on the average validation accuracy observed after training on 100 tasks. Full detail about the hyper-parameter search is described in appendix A.1.

For the selected hyper-parameter configuration, we plot the highest online test accuracy achieved for each task in Figure 5. We observe that overall, our proposed methods Interpolate and Interpolate+ReDo, consistently maintain competitive performance as other baselines. This shows that resetting the active parts of the model can also lead to improved plasticity across different distribution shifts and architectures.

On the Noisy and Permuted CIFAR10 settings, all methods result in almost identical performance except Reinit (full). On Shuffled CIFAR10 with MLP, Interpolate (active) results in the best final test accuracy. However, on Permuted CIFAR10 with MLP, ReDo outperforms other methods by a small margin. Although no single approach consistently excels in every context, both Interpolate and Interpolate+ReDo result in strong competitive performance. This highlights that resetting active neurons can be just as useful as resetting inactive ones in maintaining plasticity in online learning.

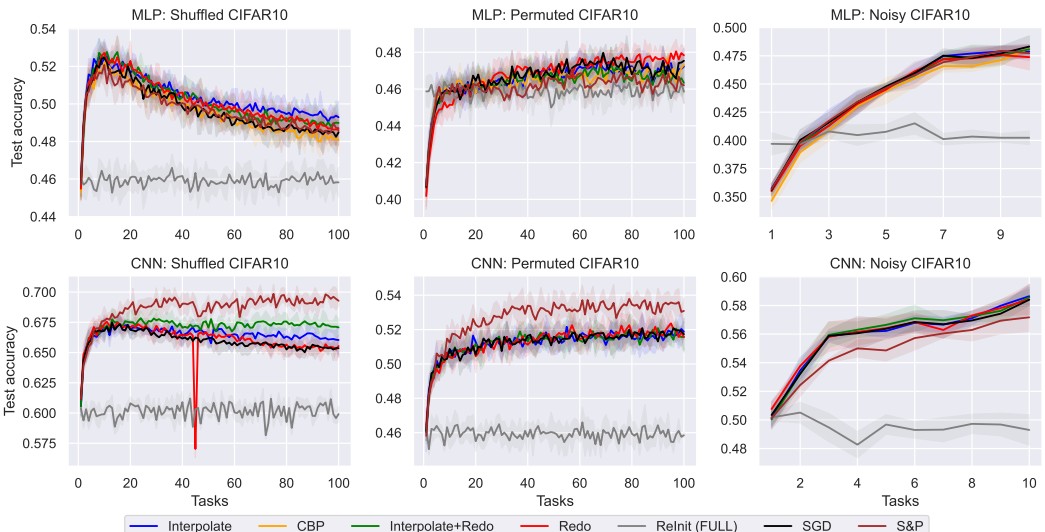

Figure 5: Comparing online test accuracy for different plasticity baselines with our proposed reset function Interpolate and Interpolate+ReDo. The best setup were obtained after an exhaustive hyper-parameter search. Overall, Interpolate and Interpolate+ReDo, consistently maintain competitive performance suggesting that resetting active neurons can also help maintain plasticity contrary to earlier assumptions.

## 4.5 LIMITATIONS

Our experiments have shown that resetting the active neurons using Interpolate can address plasticity loss in MLP and CNN with a fixed compute budget for each task. However, it raises interesting questions on its applicability to larger architectures such as Transformers (Vaswani et al., 2017). While research on plasticity in large language models is still limited, model merging has shown great promise in improving generalization in such models (Lawson & Qureshi, 2024; Verma & Elbayad, 2024; Ye et al., 2023), which suggests that resetting functions like Interpolate could be useful in this context.

While we primarily focused on CIFAR10, following existing works that have explored plasticity loss (Lyle et al., 2024; Lewandowski et al., 2024b), we have evaluated our method and baselines on different distribution shifts. This encourages further investigation into the effectiveness of our method on more realistic datasets with natural distribution shifts, such as CLoc (Cai et al., 2021).

## 5 CONCLUSION

This study provides an empirical analysis of reset-based techniques with various utility functions to address plasticity loss. Our findings challenge previous assumptions by demonstrating that resetting active neurons can also improve generalization. Moreover, by leveraging properties of the loss landscape, specifically linear mode connectivity and permutation invariance, we introduce a new model merging method called Interpolate, which can act as a reset function in online learning. We conduct a comprehensive hyper-parameter search on our proposed method as well as existing baselines under various distribution shifts, demonstrating that resetting active neurons with Interpolate yields comparable generalization performance to existing baselines that focus on resetting inactive neurons.

In future work, we plan to evaluate Interpolate on more complex models such as ResNet and Transformers to explore whether resetting active neurons can also help reduce plasticity loss in these architectures. Furthermore, we are interested in exploring the combination of Interpolate with regularization- and architecture-based methods, particularly in the context of continual learning and reinforcement learning, to evaluate its potential in addressing the specific challenges of these settings.

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

## A APPENDIX

In this section, we provide additional details and extend the results of the main paper. We describe the implementation details including hyper-parameters values used in our experiments in section A.1. All experiments were executed on an NVIDIA A100 Tensor Core GPUs machine with 40 GB memory.

In all our experiments, we generate a sequence of CIFAR10 datasets split into 40,000 training examples and 10,000 validation examples. The validation set is used to select the best-performing configuration for each baseline. Unless specified in the experiment description, the default learning rate for analyses in Figure 2, Figure 3 and Figure 4 is set to 0.01 for SGD, with no L2 regularization.

For all experiments on MLP and CNN, we used batch size of 128. Each seed ran a different randomly generated task sequence. All experiments were run in JAX (Bradbury et al., 2018), parallelized over seeds.

### A.1 TRAINING SETUP AND HYPER-PARAMETERS DETAILS

Table 1: Dataset details

| Dataset | Train set | Validation set |
|---------|-----------|----------------|
| CIFAR10 | 40K | 10K |
| CIFAR100 | 40K | 10K |

In Table 1 and Table 2, we provide a summary of datasets and models used in our experiments. We do not use any type of normalization layer in our MLP and CNN experiments.

Table 2: Model details

| Model | Number of parameters |
|-------|---------------------|
| MLP | 0.4M |
| CNN | 39K |
| ResNet18 | 11M |

The 2D contour plots in Figure 1 was obtained using loss surface visualization tool from Li et al. (2018). We compute loss by taking an average over $40$ batches ($40 \times 128/40k$ training samples) for

the loss function and computed on $100 \times 100$ models. The surface corresponds to seed 1 of Task 2 in Shuffled CIFAR10 with MLP such that *Init* is the location of model parameters on Task 2 surface after training on Task 1. After training for few epochs with SGD optimizer, once the model reaches state A, we create two copies of the model. We apply Interpolate on the second copy to obtain new location C and resume training on both copies until convergence. Additional contour plots of Test error surfaces on single task of CIFAR10 dataset are shown in Figure 6 again indicating that training from Interpolated point can result in discovery of a better generalizable region.

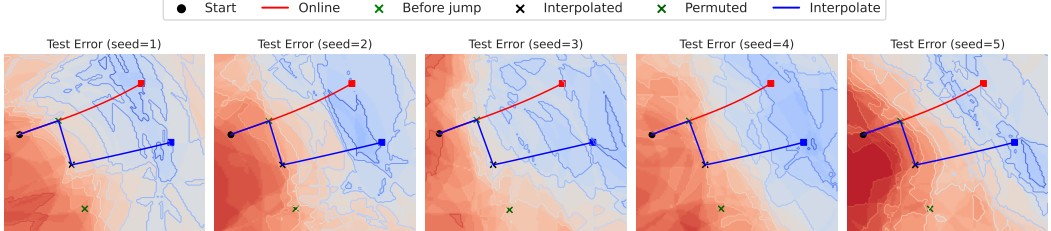

Figure 6: The 2D contour plots of Test error surfaces for 5 seeds on single task of CIFAR10 dataset on training MLP. The resulting trajectory for training from Interpolate reset (Li et al., 2018). Training from Interpolated point resulted in discovery of a better generalizable region.

We describe the hyper-parameter grids utilized in the random search to identify the optimal configuration. All hyper-parameter searches involve exploring the best optimization setup outlined in Table 3. Additionally, we also incorporate extra hyper-parameter grids introduced by individual plasticity methods (Table 4).

Table 3: Hyper-parameter grid search for base optimizer

| Method | Parameter | Values |
|--------|-----------|--------|
| SGD | L2 Weight | $0.0, 0.01, 0.0001$ |
| | Learning Rate | $0.1, 0.01, 0.001, 0.0001$ |
| | $\beta_1$ | $0.9, 0.0$ |
| Adam | L2 Weight | $0.0, 0.01, 0.0001$ |
| | Learning Rate | $0.1, 0.01, 0.001, 0.0001, 0.00001$ |
| | $\beta_2$ | $0.99, 0.999, 0.9999$ |

## A.2 BEST-PERFORMING SETUP

For our experiments in subsection 4.4, we provide the best hyper-parameter settings for all experiments in Table 5.

Table 4: Hyper-parameter grid search for plasticity methods

| Method | Parameter | Values |
|---|---|---|
| Reinit (full) | Reset Period | $1, 5, 10, 20$ |
| CBP | Decay Rate | $0.9, 0.99, 0.999$ |
| | Maturity Threshold | $100, 1000, 10000$ |
| | Replacement Rate | $1e-3, 1e-4, 1e-5, 1e-6$ |
| ReDo | Reset Period | $1, 5, 10, 20$ |
| | Dormancy Threshold | $0.05, 0.1, 0.25, 0.5$ |
| Interpolate | Reset Period | $1, 5, 10, 20$ |
| | $k$ | $5\%, 10\%, 25\%, 50\%$ |
| Interpolate+ReDo | Reset Period | $1, 5, 10, 20$ |
| | $k$ (Interpolate) | $5\%, 10\%, 25\%, 50\%$ |
| | Dormancy Threshold (ReDo) | $0.05, 0.1, 0.25, 0.5$ |
| S&P | Noise Scale | $0.001, 0.01, 0.1, 1.0$ |
| | Shrink Weight | $0.0, 0.2, 0.4, 0.6, 0.8, 1.0$ |

Table 5: Best learning setup obtained from hyper-parameter search experiment. Unline MLP and CNN, ResNet18 experiment involved search across both SGDM and Adam optimizers. * indicates best results were obtained using Adam and corresponding value of $\beta_2$ and Lr are reported.

| Model | Data | Method | L2 | Dormancy threshold | $k$ | Reset period | $\beta$ | Lr | Noise scale | Shrink weight | Decay rate | Maturity threshold | Replacement rate |
|---|---|---|---|---|---|---|---|---|---|---|---|---|---|
| MLP | Noisy CIFAR10 | CBP | 0 | | | | 0.9 | 0.01 | | | 0.999 | 100 | 0.0001 |
| MLP | Noisy CIFAR10 | Interpolate | 0 | 0 | 0.5 | 2000 | 0 | 0.01 | | | | | |
| MLP | Noisy CIFAR10 | Interpolate+Redo | 0 | 0.5 | 0.05 | 4000 | 0 | 0.01 | | | | | |
| MLP | Noisy CIFAR10 | ReInit | 0 | | | | 0 | 0.1 | | | | | |
| MLP | Noisy CIFAR10 | Redo | 0 | 0.05 | 0 | 1000 | 0 | 0.01 | | | | | |
| MLP | Noisy CIFAR10 | SGD | 0 | | | | 0.9 | 0.1 | | | | | |
| MLP | Noisy CIFAR10 | S&P | 0 | | | | 0 | 0.01 | 0.001 | 0.2 | | | |
| MLP | Permuted CIFAR10 | CBP | 0.01 | | | | 0 | 0.01 | | | 0.9 | 100 | 1e-06 |
| MLP | Permuted CIFAR10 | Interpolate | 0.0001 | 0 | 0.1 | 2000 | 0 | 0.01 | | | | | |
| MLP | Permuted CIFAR10 | Interpolate+Redo | 0.01 | 0.05 | 0.05 | 1000 | 0.9 | 0.01 | | | | | |
| MLP | Permuted CIFAR10 | ReInit | 0.01 | | | | 0.9 | 0.1 | | | | | |
| MLP | Permuted CIFAR10 | Redo | 0 | 0.1 | 0 | 200 | 0 | 0.01 | | | | | |
| MLP | Permuted CIFAR10 | SGD | 0.0001 | | | | 0 | 0.01 | | | | | |
| MLP | Permuted CIFAR10 | S&P | 0 | | | | 0 | 0.01 | 0.1 | 0.2 | | | |
| MLP | Shuffled CIFAR10 | CBP | 0.0001 | | | | 0 | 0.1 | | | 0.999 | 10000 | 1e-06 |
| MLP | Shuffled CIFAR10 | Interpolate | 0.0001 | 0 | 0.25 | 1000 | 0 | 0.1 | | | | | |
| MLP | Shuffled CIFAR10 | Interpolate+Redo | 0 | 0.1 | 0.5 | 1000 | 0.9 | 0.1 | | | | | |
| MLP | Shuffled CIFAR10 | ReInit | 0.0001 | | | | 0.9 | 0.1 | | | | | |
| MLP | Shuffled CIFAR10 | Redo | 0.01 | 0.1 | 0 | 1000 | 0 | 0.1 | | | | | |
| MLP | Shuffled CIFAR10 | SGD | 0 | | | | 0.9 | 0.1 | | | | | |
| MLP | Shuffled CIFAR10 | S&P | 0 | | | | 0 | 0.1 | 0.1 | 0.2 | | | |
| CNN | Noisy CIFAR10 | Interpolate | 0.0001 | 0 | 0.1 | 2000 | 0.9 | 0.1 | | | | | |
| CNN | Noisy CIFAR10 | Interpolate+Redo | 0.01 | 0.5 | 0.25 | 4000 | 0.9 | 0.1 | | | | | |
| CNN | Noisy CIFAR10 | CBP | 0.0001 | | | | 0.9 | 0.1 | | | 0.999 | 10000 | 0.000001 |
| CNN | Noisy CIFAR10 | ReInit | 0 | | | | 0.9 | 0.1 | | | | | |
| CNN | Noisy CIFAR10 | Redo | 0.01 | 0.05 | 0 | 200 | 0 | 0.1 | | | | | |
| CNN | Noisy CIFAR10 | SGD | 0 | | | | 0.9 | 0.1 | | | | | |
| CNN | Noisy CIFAR10 | S&P | 0 | | | | 0.9 | 0.1 | 0.01 | 0.6 | | | |
| CNN | Permuted CIFAR10 | CBP | 0.01 | | | | 0.9 | 0.1 | | | 0.99 | 1000 | 0.001 |
| CNN | Permuted CIFAR10 | Interpolate | 0.0001 | 0 | 0.5 | 4000 | 0.9 | 0.1 | | | | | |
| CNN | Permuted CIFAR10 | Interpolate+Redo | 0 | 0.25 | 0.05 | 2000 | 0 | 0.1 | | | | | |
| CNN | Permuted CIFAR10 | ReInit | 0.01 | | | | 0 | 0.1 | | | | | |
| CNN | Permuted CIFAR10 | Redo | 0.0001 | 0.1 | 0 | 200 | 0 | 0.1 | | | | | |
| CNN | Permuted CIFAR10 | SGD | 0 | | | | 0 | 0.1 | | | | | |
| CNN | Permuted CIFAR10 | S&P | 0 | | | | 0 | 0.1 | 0.01 | 0.4 | | | |
| CNN | Shuffled CIFAR10 | CBP | 0.01 | | | | 0.9 | 0.1 | | | 0.999 | 100 | 0.000001 |
| CNN | Shuffled CIFAR10 | Interpolate | 0 | 0 | 0.05 | 1000 | 0.9 | 0.1 | | | | | |
| CNN | Shuffled CIFAR10 | Interpolate+Redo | 0.01 | 0.05 | 0.05 | 1000 | 0.9 | 0.1 | | | | | |
| CNN | Shuffled CIFAR10 | ReInit | 0.0001 | | | | 0.9 | 0.1 | | | | | |
| CNN | Shuffled CIFAR10 | Redo | 0 | 0.1 | 0 | 1000 | 0.9 | 0.1 | | | | | |
| CNN | Shuffled CIFAR10 | SGD | 0.0001 | | | | 0 | 0.1 | | | | | |
| CNN | Shuffled CIFAR10 | S&P | 0 | | | | 0 | 0.1 | 1 | 0.2 | | | |
| ResNet18 | Noisy CIFAR100 | CBP | 0.0001 | | | | 0.999 | 0.0001* | | | 0.999 | 10000 | 0.000001 |
| ResNet18 | Noisy CIFAR100 | Interpolate | 0.0001 | 0 | 0.1 | 10000 | 0 | 0.1 | | | | | |
| ResNet18 | Noisy CIFAR100 | Interpolate+Redo | 0.0001 | 0.02 | 0.2 | 2000 | 0.99 | 0.001* | | | | | |
| ResNet18 | Noisy CIFAR100 | ReInit | 0.0 | | | | 0.999 | 0.001* | | | | | |
| ResNet18 | Noisy CIFAR100 | Redo | 0.001 | 0.02 | | 400 | 0.999 | 0.001* | | | | | |
| ResNet18 | Noisy CIFAR100 | SGD | 0.0001 | | | | 0.999 | 0.0001* | | | | | |
| ResNet18 | Permuted CIFAR100 | CBP | 0.01 | | | | 0.9 | 0.1 | | | 0.99 | 1000 | 0.001 |
| ResNet18 | Permuted CIFAR100 | Interpolate | 0.01 | 0 | 0.05 | 10000 | 0.9 | 0.1 | | | | | |
| ResNet18 | Permuted CIFAR100 | Interpolate+Redo | 0 | 0.05 | 0.02 | 2000 | 0.9 | 0.1 | | | | | |
| ResNet18 | Permuted CIFAR100 | ReInit | 0.0 | | | | 0.999 | 0.001* | | | | | |
| ResNet18 | Permuted CIFAR100 | SGD | 0.01 | | | | 0.9 | 0.01 | | | | | |
| ResNet18 | Permuted CIFAR100 | Redo | 0.01 | 0.5 | | 200 | 0.9 | 0.1 | | | | | |
| ResNet18 | Shuffled CIFAR100 | CBP | 0.01 | | | | 0.9 | 0.1 | | | 0.999 | 100 | 0.000001 |
| ResNet18 | Shuffled CIFAR100 | Interpolate | 0.01 | 0 | 0.1 | 1000 | 0.9 | 0.1 | | | | | |
| ResNet18 | Shuffled CIFAR100 | Interpolate+Redo | 0.01 | 0.05 | 0.1 | 10000 | 0.9 | 0.1 | | | | | |
| ResNet18 | Shuffled CIFAR100 | ReInit | 0.0001 | | | | 0.9999 | 0.001* | | | | | |
| ResNet18 | Shuffled CIFAR100 | Redo | 0.01 | 0.1 | | 10000 | 0.9 | 0.1 | | | | | |
| ResNet18 | Shuffled CIFAR100 | SGD | 0.01 | | | | 0.9 | 0.1 | | | | | |

### A.3 ADDITIONAL RESULTS

#### A.3.1 COMPARING WITH BASELINES ON RESNET

For ResNet-18 (He et al., 2016), each task consisted of 20,000 gradient steps also with batch size 256. We conduct a hyper-parameter search for training ResNet18 on all three types of non-stationary setting similar to subsection 4.4. We use CIFAR100 dataset. In Figure 7, we observe that either Interpolate or Interpolate+ReDo, exhibit competitive/better performance suggesting that resetting active neurons can also help maintain plasticity.

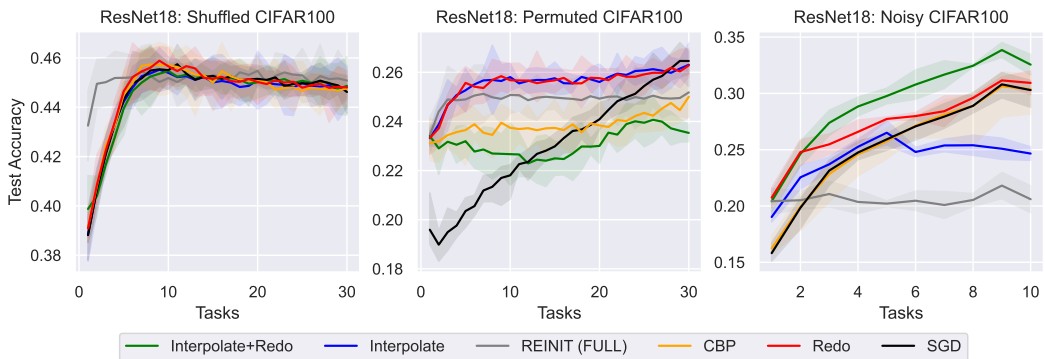

Figure 7: Comparing online test accuracy for different plasticity baselines with Interpolate and Interpolate+ReDo on training ResNet18 using CIFAR100 dataset. The best setup were obtained after an exhaustive hyper-parameter search. Either Interpolate or Interpolate+ReDo, exhibit competitive/better performance suggesting that resetting active neurons can also help maintain plasticity.

#### A.3.2 LARGER NUMBER OF TASKS

In Figure 8, we compare online test accuracy of Interpolate and Interpolate+ReDo with baselines for training on larger number of tasks (400) on Permuted CIFAR10 and Permuted MNIST. In both cases, Interpolate and Interpolate+ReDo consistently maintained similar performance as Redo.

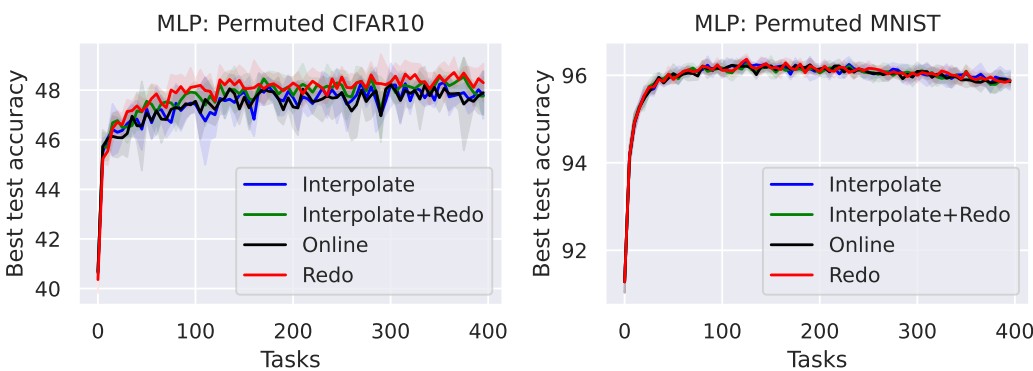

Figure 8: Evaluating online test accuracy of Interpolate and Interpolate+ReDo for larger number of tasks on Permuted CIFAR10 and Permuted MNIST. Overall, Interpolate and Interpolate+ReDo, consistently maintain similar performance as Redo again suggesting that resetting active neurons can also help maintain plasticity contrary to earlier assumptions.

#### A.3.3 WITH ADAM OPTIMIZER

In Figure 9, we conduct an ablation study and evaluate Interpolate and Interpolate+ReDo using Adam as base optimizer. Details for the hyper-parameter search is given in Table 3. Interpolate performs best on Noisy CIFAR10 and maintain similar performance as Redo on Shuffled CIFAR10.

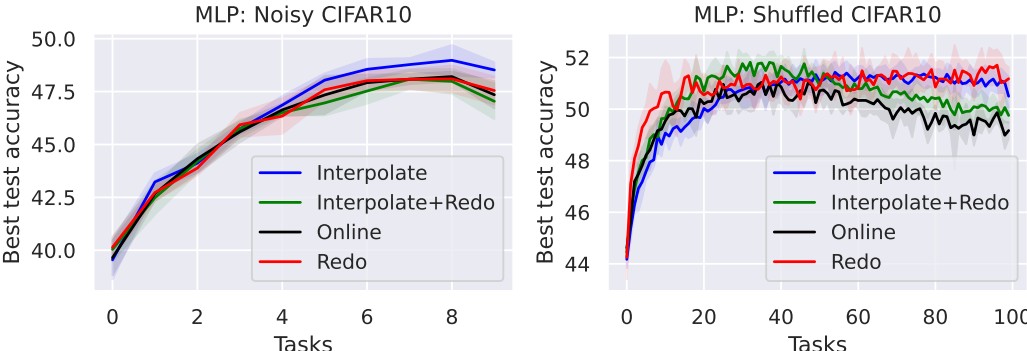

Figure 9: Evaluating online test accuracy of Interpolate and Interpolate+ReDo with Adam optimizer. Interpolate performs best on Noisy CIFAR10 and maintain similar performance as Redo on Shuffled CIFAR10.

### A.3.4 HIGHER COMPUTE BUDGET PER TASK

In Figure 10, we compare online test accuracy of Interpolate and Interpolate+ReDo with baselines for training on larger number of epochs per task (100) on Permuted CIFAR10 and Shuffled CIFAR100. While Redo slightly performs better on Permuted CIFAR10, Interpolate performs better on Shuffled CIFAR10.

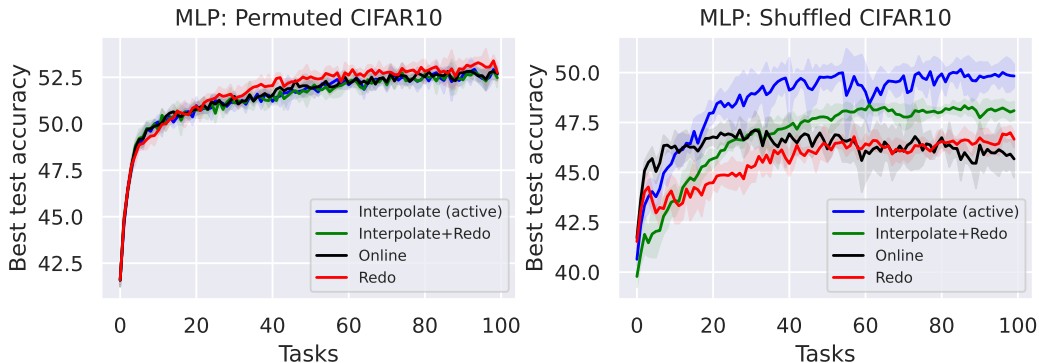

Figure 10: Evaluating online test accuracy of Interpolate and Interpolate+ReDo for larger number of epochs (100) per task on Permuted CIFAR10 and Shuffled CIFAR10. While Redo slightly performs better on Permuted CIFAR10, Interpolate clearly performs best on Shuffled CIFAR10.

### A.3.5 CONVEX COMBINATIONS

Here, we define $\theta_{\texttt{reset}}$ as convex combination of $\theta_{\texttt{perm}}$ and $\theta$:

$$\theta_{\texttt{reset}} = w\theta_{\texttt{perm}} + (1 - w)\theta \,,$$

where $w$ is the interpolate weight. We vary $w$ and train an MLP on Shuffled CIFAR10 for 100 tasks. We plot the results in Figure 11 and observe that while with larger learning rate, varying $w$ has minimal effect on overall performance, with smaller learning rate, $w = 0.6$ works best in maintain plasticity and $w = 0.9$ diverges on later tasks.

### A.3.6 MULTIPLE PERMUTATIONS

Here, we define $\theta_{\texttt{reset}}$ as average across multiple $\theta_{\texttt{perm}}$ generated, i.e.,

$$\theta_{\texttt{reset}} = \frac{1}{t+1}(\theta + \sum_{i=1}^{t} \theta_{\texttt{perm}-i}) \,.$$

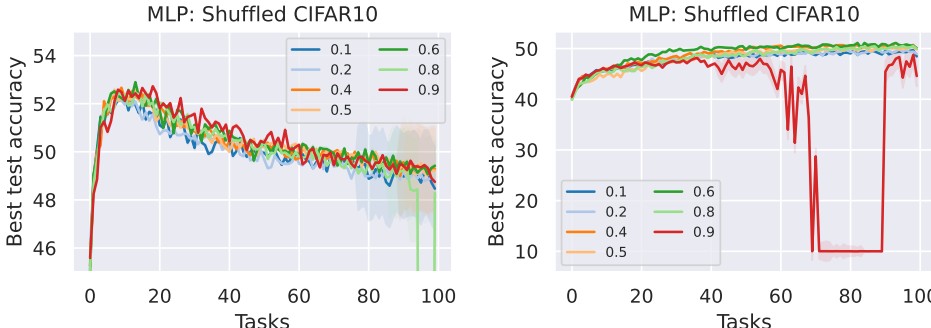

Figure 11: Evaluating online test accuracy of Interpolate on Shuffled CIFAR10 across different interpolate weights with learning rates: (i) 0.1 (ii) 0.01. We observe that while with larger learning rate, varying $w$ has minimal effect on overall performance, with smaller learning rate, $w = 0.6$ works best in maintain plasticity and $w = 0.9$ diverges on later tasks.

We vary $n$ and train an MLP on Shuffled CIFAR10 for 100 tasks. We plot the results in Figure 12 (left) and observe that $n$ has minimal impact on overall performance.

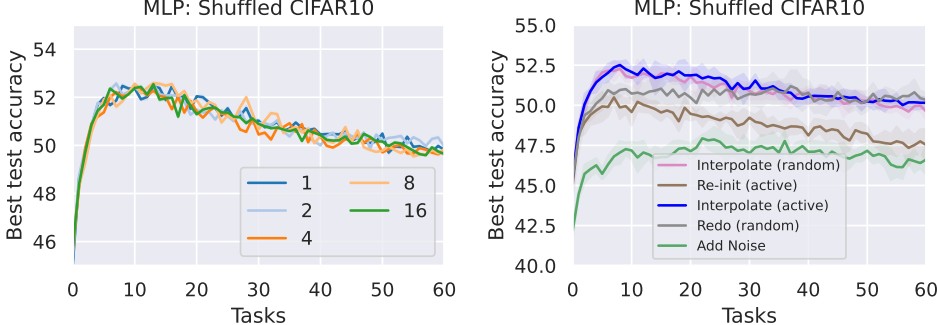

Figure 12: Evaluating online test accuracy of Interpolate on Shuffled CIFAR10: (i) across different number of permutations where, we observe that it has minimal impact on overall performance; (ii) with additional baselines involving *random* selection of neurons, *re-init*ialization, adding *noise*. Interpolate with both active and random neurons selection perform similar. Redo with random neurons selection also results in competitive performance on later tasks whereas both re-initializing active neurons and adding noise exhibit worse performance.

### A.3.7 RANDOM SELECTION

In this experiment, we add more baselines: (i) *random* selection of neurons, (ii) *re-init*ialization, (iii) adding *noise*. Figure 12 (right) shows that Interpolate with both active and random neurons selection results in similar performance. Redo with random neurons selection also results in competitive performance on later tasks whereas both re-initializing active neurons and adding noise exhibit worse performance.

### A.3.8 JUMP IN TRAINING LOSS VS ACTIVATION SCORE

Similar to Figure 2, we compare generalization performance and jump in training loss for random noise and Interpolate reset functions with increasing total activation score of randomly selected neurons. In Figure 13, we observe that Interpolate results in relatively more efficient adaptation to new tasks, while random noise can introduce instability and performance loss when applied to more active neurons.

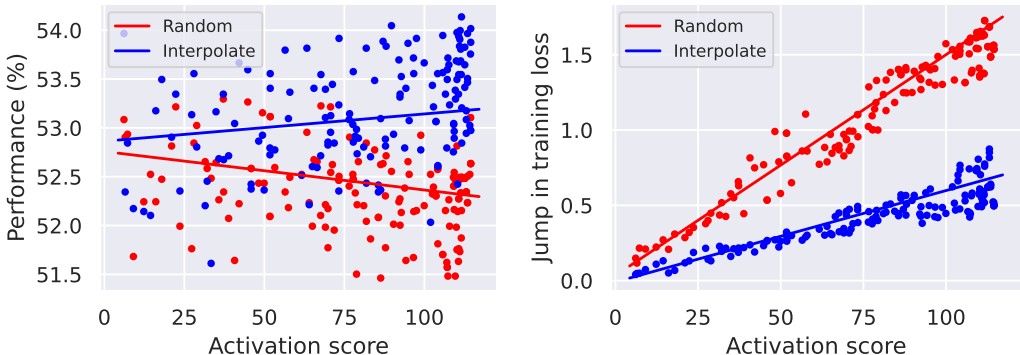

Figure 13: Comparing generalization performance (left) and jump in training loss (right) for random noise and Interpolate reset functions for increasing total activation score of randomly selected neurons. Similar to Figure 2, Interpolate results in relatively more efficient adaptation to new tasks, while random noise can introduce instability and performance loss when applied to more active neurons.

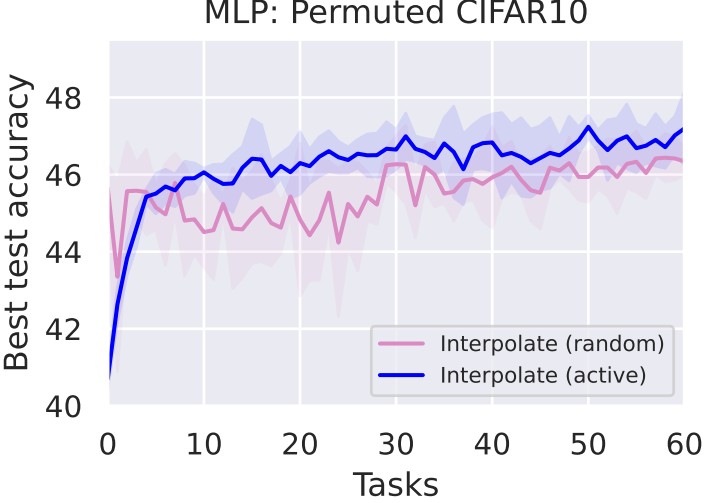

Figure 14: Evaluating online test accuracy of Interpolate on Permuted CIFAR10 for comparing Interpolate with *random* selection of neurons on Permuted CIFAR10. Interpolate with both active and random neurons selection perform worse.

### A.3.9  SENSITIVITY ANALYSIS

In this section, we provide a brief sensitivity analysis of Interpolate and Interpolate+ReDo for different values of $k$ and dormancy threshold on training CNN using Permuted CIFAR10 and Shuffled CIFAR10 dataset. Figure 15 shows that in case of Inteprolate, a higher value of $k$ works better in terms of overall performance. While there's no clear trend in case of Interpolate+Redo as different combinations work well, a higher dormancy threshold results in worse performance.

### A.3.10  OTHER METRICS OBSERVED USING BEST PERFORMING SETUP

In this section we plot other metrics including final train accuracy and weight norm obtained for the best performing hyparmeter configurations.

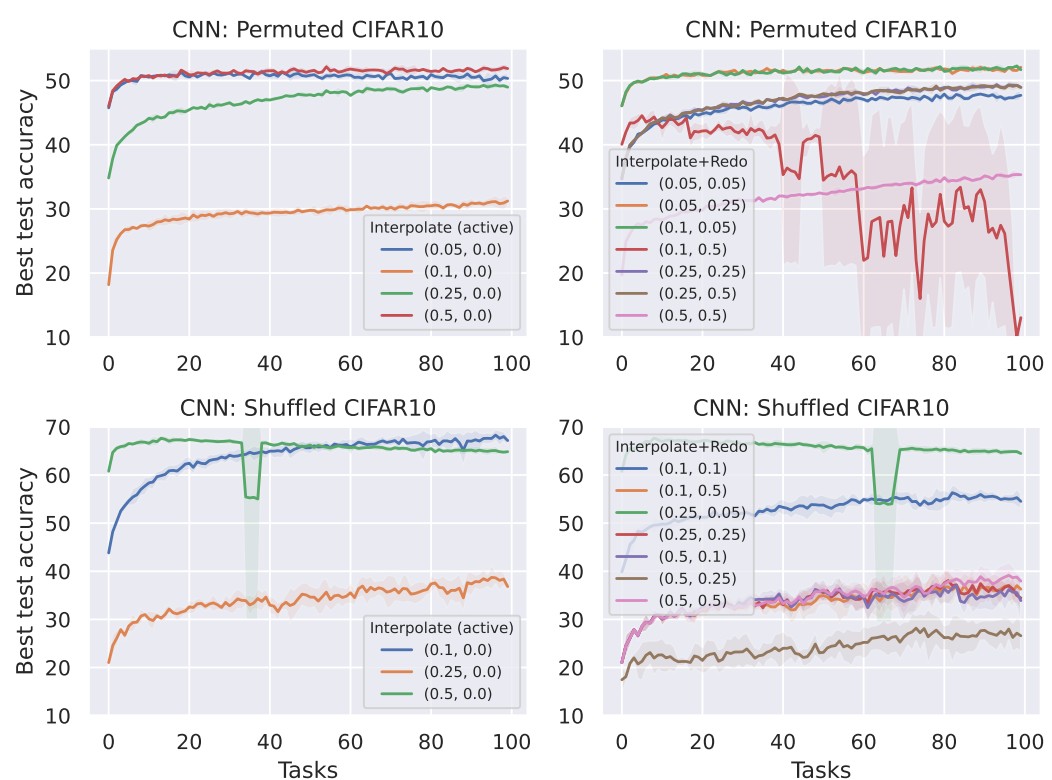

Figure 15: Comparing online test accuracy for different values of $k$ with Interpolate and ($k$, dormancy threshold) with Interpolate+ReDo on training CNN using Permuted CIFAR10 and Shuffled CIFAR10 dataset after the hyper-parameter search. Higher value of $k$ works better for Interpolate. While there's no clear trend in case of Interpolate+Redo as different combinations work well for both benchmarks, a higher dormancy threshold results in worse performance.

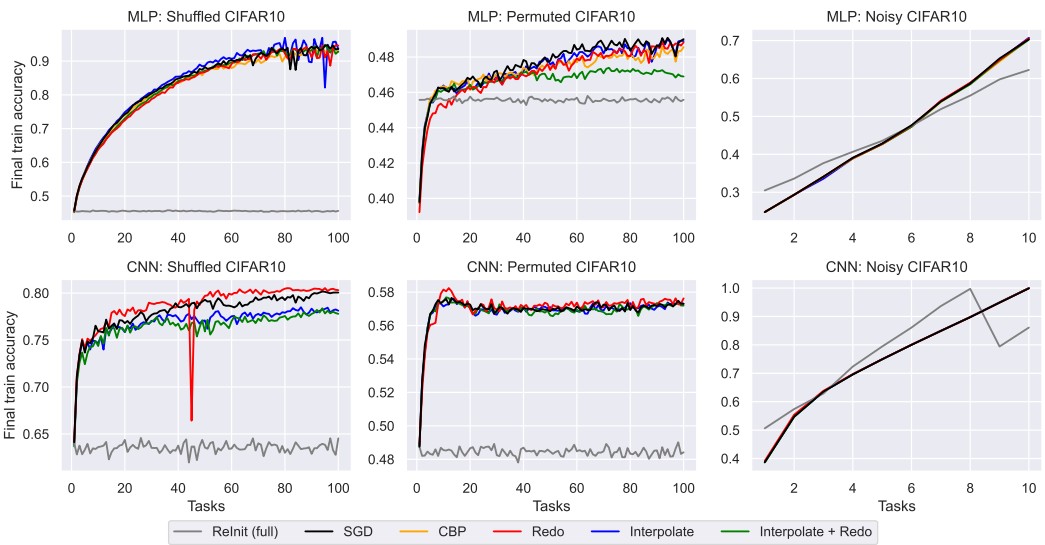

Figure 16: Comparing online train accuracy for different plasticity baselines with our proposed reset function Interpolate and Interpolate+ReDo after the hyper-parameter search. Overall, Interpolate and Interpolate+ReDo, consistently maintain similar performance on all settings except CNN+Shuffled CIFAR10 where ReDo performs best.

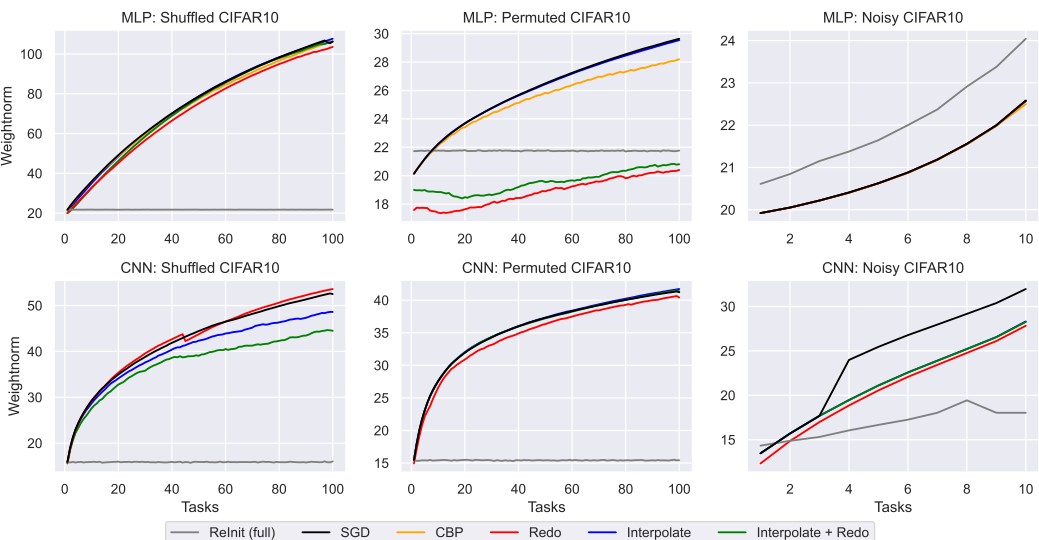

Figure 17: Comparing weight norm for different plasticity baselines with our proposed reset function Interpolate and Interpolate+ReDo after the hyper-parameter search. While all palsticity methods result in similar increase in the weight norm, the only exception occurs with MLP+Permuted CIFAR10 where ReDo and Interpolate+ReDo maintains a smaller weight norm under their best configurations.

