# OpenReview forum: "Interpolate: How Resetting Active Neurons can also improve Generalizability in Online Learning"
_ICLR.cc/2025/Conference — Submitted to ICLR 2025_

### Official Review · Reviewer_QsVM · 2024-10-31

**Soundness:** 3
**Presentation:** 3
**Contribution:** 3
**Rating:** 5
**Confidence:** 5

**Summary:**

The paper proposes a reset-inspired method, referred to as Interpolate, for maintaining plasticity and increasing generalization in online learning. Interpolate periodically updates a neural network's parameters by interpolating between the the network parameters and a functionally equivalent permutation, where the permutation permutes a subset of `"active’’ or high-utility neurons. The method is evaluated against competitor reset-based continual learning algorithms, Continual Backdrop and ReDO, on a series of continual learning problems on the CIFAR10 dataset. Results show that interpolating network parameters with a random (functionally equivalent) permutation of the most active neurons is competitive with Continual Backdrop and ReDO.

**Strengths:**

- The paper considers a completely new paradigm of resetting active or high-utility neurone, which runs counter to existing reset-based methods that reset dormant or low-utility neurone. The results of this paper are surprisingly positive, counter to conventional expectations of neuron dormancy being a driving factor of plasticity loss. For this conjunction of reasons, this is a worthwhile contribution.
- The paper is generally well written and clear with its stated contributions.

**Weaknesses:**

- The claim that resetting active neurons improves generalization is not entirely supported by the results. The algorithm Interpolate interpolates between the network’s parameters and a functionally equivalent permutation — this is not ``full’’ resetting of active neurons.
- Not all of the experiments in section 4.4 exhibit plasticity loss. Specifically for Permuted CIFAR10 and Noisy CIFAR10 we see that SGD attains greater test accuracy over time. However, plasticity loss is the phenomenon in which baselines like SGD experience diminishing performance as the network is faced with more tasks, but for the aforementioned experiments we do not see this phenomenon. To validate that Interpolate mitigates plasticity loss, we should be seeing this phenomenon arising in a baseline SGD model. I would recommend increasing the number of tasks and/or evaluating additional experiments such as Permuted MNIST or Continual Imagenet (see Dohare et al. or Kumar et al.).

**Questions:**

- Can you formally define `"functionally equivalent parameters" for completeness and in order to be precise.
- Out of curiosity, have the author(s) considered other convex combinations of $\theta_{\text{perm}}$ and $\theta$, rather than a 0.5-0.5 combination? Perhaps more frequent and gradual interpolations may be performant.
- I understand that Interpolate captures the spirit of resetting active neurone, but I feel that it is more of a model merging technique, merging the network’s parameters with a functionally equivalent permutation, and does not perform ``full’’ resets. This does not square entirely with the title, abstract, and introduction which claim that your algorithm Interpolate and experiments involve resetting active neurons. Could the paper be worded differently in a way that makes it clear that the reset-technique is really a model merging technique? Reading up to section 3.1 I was expecting to see resets in the flavour of CBP and ReDO, with an additional model merging algorithm as a secondary contribution, but instead the "resets of active neurons" are instead interpolations between active neurons.
- In section 4.1, precisely what is reset by random noise? What is the noise scale and are neurons reset according to the prior distribution used for initialization?
- Given that dormancy continues to increase in Figure 2 and does not plateau, it would be useful to see how these statistics (test accuracy, dormancy, etc.) evolve over more tasks.
- Can some of these experiments be replicated with the Adam optimizer?

---

> ### Author Response · Authors · 2024-11-23
>
> We’d like to thank the reviewer for the positive evaluation of our paper about the new paradigm, our contribution, overall clarity and results presented in our method. We appreciate the suggestions and provide our responses below:
>
> **Reviewer’s Comment: “The algorithm Interpolate … is not ``full’’ resetting of active neurons. … Could the paper be worded differently in a way that makes it clear that the reset-technique is really a model merging technique”**
>
> **Response:** Thank you for pointing it out, we believe that using the term “reset” might have caused confusion. To address this, we have added a clarification in the introduction, specifying that by ‘reset,’ we specifically mean re-locating the parameters to a different point on the loss landscape. This definition of resetting includes any form of re-locating, including model merging.”
>
> ----
>
>
> **Reviewer’s Comment: “Not all of the experiments in section 4.4 exhibit plasticity loss. … I would recommend increasing the number of tasks and/or evaluating additional experiments such as Permuted MNIST”**
>
> **Response:**
> Please note that we use L2 regularization in our optimization setup by default which itself is a good baseline method that addresses plasticity loss.
> - Noisy CIFAR10: The reason we do not see plasticity loss in Noisy CIFAR10 is because as the label noise decreases for subsequent tasks, the model should become more confident about the dataset and improve its performance.
> - Permuted CIFAR10: We’ve used 10 epochs in Permuted CIFAR10 which underfits on individual tasks and therefore a significant forward transfer occurs as the model encounters new tasks.
> - Based on the reviewer's recommendation, we have added experiments with a higher number of tasks to verify if plasticity loss occurs at later stages. We have also added Permute MNIST in these experiments described in Appendix A.3.2. We observed that both Interpolate and Interpolate+ReDo consistently maintained similar performance as Redo
>
>
> ----
>
>
> **Reviewer’s Comment: “Can you formally define `"functionally equivalent parameters" for completeness and in order to be precise.”**
>
> **Response:** Thank you for your suggestion, we have updated section 3.2 accordingly to define “functionally equivalent parameters”.
>
>
> ----
>
>
> **Reviewer’s Comment: “have the author(s) considered other convex combinations of $\theta_{\text{perm}}$ and $\theta$”**
>
> **Response:** Thank you for the suggestion. We’ve added ablation with other convex combinations A.3.5 where we evaluate our method across different interpolate weights with learning rates: (i) 0.1 (ii) 0.01. We observe that while with larger learning rate, varying $w$ has minimal effect on overall performance, with smaller learning rate, $w=0.6$ works best in maintaining plasticity and $w=0.9$ diverges on later tasks
>
>
> ----
>
>
> **Reviewer’s Comment: “what is reset by random noise?”**
>
> **Response:** Thank you for pointing it out. We'd like to clarify that the neurons are reset using the Lecun normal initializer. We’ve added it in the paper.
>
> ----
>
>
> **Reviewer’s Comment: “dormancy continues to increase in Figure 2 and does not plateau, it would be useful to see how these statistics evolve over more tasks.
> ”**
>
> **Response:** Thank you for the suggestion. To validate whether increasing dormancy might hurt the performance on future tasks, we’ve also added results with more tasks in Appendix A.3.2 showing that it does not play a huge role in overall behavior and Interpolate still shows similar performance as other baselines.
>
> ----
>
>
>
> **Reviewer’s Comment: “Can some of these experiments be replicated with the Adam optimizer?”**
>
> **Response:** Thank you for this suggestion, we’ve conducted a similar hyper-parameter search experiment using Adam optimizer instead of SGD on Shuffled CIFAR10 and Noisy CIFAR10. We’ve added the results in Appendix A.3.3. We observed that Interpolate performs best on Noisy CIFAR10 but maintains similar performance as Redo on ShuffledCIFAR10.
>
>
> ----
>
>
> **We greatly appreciate this review, whose feedback will help us improve our work. We hope that the above responses address the reviewer’s concerns. If the reviewer feels the same, it would be appreciated if this could be reflected through an updated score. If the reviewer has any remaining questions, please do not hesitate to ask, and we will do our best to answer them as soon as possible.**

---

> > ### Author Response · Authors · 2024-11-28
> >
> > Dear Reviewer QsVM, we hope that you've had a chance to read our responses and clarification. We would greatly appreciate it if you could confirm whether our updates have addressed your concerns and, if possible, support our work by considering an increased score. If the reviewer has any remaining questions, please do not hesitate to ask, and we will do our best to answer them as soon as possible.

---

> > > ### Comment · Reviewer_QsVM · 2024-12-03
> > >
> > > For the most part these revisions address my concerns. However, I maintain my score of 5 as I agree with many of the concerns presented by the other reviewers.
> > >
> > > My key concern, that is tangentially echoed by other reviewers, is that interpolation is not necessarily resetting of neurons. I appreciate the qualification that resetting is to encompass any “relocation of parameters elsewhere on the loss surface”. But with all due respect, I would disagree with this broad definition as the plasticity community has generally defined resetting as reinitializing parameters according to the original distribution used to initialize the network weights, or via some comparable distribution. To position this paper as going against the paradigm of “dormant neurons being the only useful neurons to reset” is somewhat misleading given that interpolation is not resetting in the spirit that I have laid out, and I don’t believe that a brief redefinition of neuron resets is appropriate.
> > >
> > > I actually enjoyed reading this paper and found it a worthwhile contribution, but I do believe that the story that is being told could be improved, and the necessary revision is beyond the scope of a two-week rebuttal. Rather than positioning this paper within the paradigm of neuron resets, I would instead argue that the model merging technique falls under a new class of interventions for mitigating plasticity loss — and this should be the story presented by the authors.

---

### Official Review · Reviewer_siNw · 2024-11-02

**Soundness:** 1
**Presentation:** 2
**Contribution:** 3
**Rating:** 3
**Confidence:** 3

**Summary:**

This work shows a new empirical insight to address the loss of plasticity, a tendency of neural networks (NNs) to lose their ability to adapt to new tasks in non-stationary settings. The authors show in their experiments that even resetting the most active parameters could be an alternative way to address the loss of plasticity, in contrast to previous studies that reset the inactive neurons. Two main results are highlighted. First, the results comparing the different utility functions for selecting which neurons to reset. Second, the performance of the proposed Interpolate method, which exploits the model merging using permutation invariance. Based on these results, the authors present experimental results across different distribution shift scenarios on simple MLP and CNN settings, which show comparable performance to existing baselines.

**Strengths:**

Providing a empirical evidence that resetting the active neurons can help NNs overcome the loss of plasticity, which is a new insight to this field of study, is a strength of this work.

Section 4.1 shows the effect of reset function, specifically comparing the Interpolate and random noise (same random distribution as the NN initialization). As the authors mentioned, many prior works utilizes random noise same to the initialization to reset, and the comparing this to the proposed Interpolate is a strength of this work.

Section 4.2 then presents the effect of utility function by comparing several candidates of choices for which neurons to reset. While this result does not show a definite trend to explain the effect of utility function, identifying that the choice of utility function that chooses the active neurons is a strength of this work.

**Weaknesses:**

Overall, I don't expect Interpolate or Interpolate+Redo to beat other resetting methods, for example in Figure 4, where the authors suggest that Interpolate-based methods maintain competitive performance, which I acknowledge that resetting non-dormant neurons is indeed a new insight. However, I would argue that this perspective should be accompanied by more extensive analysis (either empirical or theoretical). Below are some of the examples that I think should be not left undiscussed.

1. How is it guaranteed that the merged model itself provides an appropriate place for reset? While I see that the last paragraph (lines 194-197) in Section 3.2 tries to give an explanation for this, however I see no empirical evidence to support it. For example, comparison between $\theta$, $\theta_\text{perm}$, $\theta_\text{reset}$, $\theta_\text{reset}+\epsilon$, (where $\epsilon$ is random perturbation of the model), or averageing more than two permuted model ($\frac{\theta + \sum_{i=1}^{N-1}{\theta^i_\text{perm}}}{N}$) could aid the authors' claim for using $\theta_\text{reset}$. I would like to see an additional experimental results on this point, either based on my suggestions or on the authors' own.
2. I think that lines 197-199 is the one of the main hypotheses that the authors make. So I would expect a thorough empirical evidence to support this claim, which is not included in the main text.
3. (Closely related to the above 2.) I assume Figure 2 is objected to in order to analyze the advantage of resetting non-dormant neurons compared to other methods. While the authors conclude that there is an ambiguity of the metrics to explain the results, I think these metrics should be more thoroughly designed to expect such effects. Authors could design a different metric to logically/effectively support the result, or measure these metrics in extensive ablation experiments as in the setting of Figure 3 (i.e. controlling $k$). Maybe the difference of the loss landscape or the changes of activations/representations before and after the reset could be a possible candidate to measure.
4. The phenomenon of the loss of plasticity is also highlighted in the field of reinforcement learning. And among the references cited by the authors, there are papers that study the effect of resetting in reinforcement learning. I think this work could be more strengthened by adding experiments in the reinforcement learning setting as in [1, 2, 3].

[1] Maximilian Igl, Gregory Farquhar, Jelena Luketina, Wendelin Boehmer, and Shimon Whiteson. Transient non-stationarity and generalisation in deep reinforcement learning. arXiv preprint arXiv:2006.05826, 2020.

[2] Evgenii Nikishin, Max Schwarzer, Pierluca D’Oro, Pierre-Luc Bacon, and Aaron Courville. The primacy bias in deep reinforcement learning. In International conference on machine learning, 2022.

[3] Ghada Sokar, Rishabh Agarwal, Pablo Samuel Castro, and Utku Evci. The dormant neuron phenomenon in deep reinforcement learning. In International Conference on Machine Learning, 2023.

**Questions:**

1. I think the results of Section 4.1 could be seen less (statistically) significant to some readers, especially the left subfigure. The correlation seems weak, and even my statement could not be guaranteed with only one seed. Can
2. I'm aware that prior works like [1, 2] uses the experimental settings in similar scale, but I'm curious that can the effect that the authors present also be seen on more larger/complex model sizes/architectures?
3. Can the authors clarify when is the point the model undergoes reset for the experiments in Section 4.2? I couldn't find this setting only for Section 4.2.
4. May be the authors could try different reset functions. I think the main contribution of this paper is selecting the non-dormant neurons to reset. If this is robust across diverse reset functions, this could be a certain strength. Based on the taxonomy based on [3], while the reset method the authors are using is re-randomization, there exists other methods; re-initializing [4, 5], or using checkpoints [6].



[1] Alex Lewandowski, Haruto Tanaka, Dale Schuurmans, and Marlos C. Machado. Directions of curvature as an explanation for loss of plasticity, 2024

[2] Clare Lyle, Zeyu Zheng, Khimya Khetarpal, Hado van Hasselt, Razvan Pascanu, James Martens, and Will Dabney. Disentangling the causes of plasticity loss in neural networks. arXiv preprint arXiv:2402.18762, 2024.

[3] Zhang, Chiyuan, Samy Bengio, and Yoram Singer. Are all layers created equal?. Journal of Machine Learning Research 23.67 (2022): 1-28.

[4] Evgenii Nikishin, Max Schwarzer, Pierluca D’Oro, Pierre-Luc Bacon, and Aaron Courville. The primacy bias in deep reinforcement learning. In International conference on machine learning, 2022.

[5] Oh, Jaehoon, et al. ReFine: Re-randomization before Fine-tuning for Cross-domain Few-shot Learning. Proceedings of the 31st ACM International Conference on Information & Knowledge Management. 2022.

[6] Bae, Youngkyoung, Yeongwoo Song, and Hawoong Jeong. Stochastic Restarting to Overcome Overfitting in Neural Networks with Noisy Labels. arXiv preprint arXiv:2406.00396 (2024).

---

> ### Author Response · Authors · 2024-11-23
> **Response 1/2**
>
> We’d like to thank the reviewer for the evaluation of our paper. We’re glad that the reviewer appreciated the insights drawn from the empirical results presented in our paper. We also appreciate the suggestions and provide our responses below:
>
> **Reviewer’s Comment: “How is it guaranteed that the merged model itself provides an appropriate place for reset? … I would expect a thorough empirical evidence to support this claim… design a different metric to logically/effectively support the result”**
>
> **Response:**
> We’d like to address reviewer’s concerns as following:
> - While there are several works (mentioned in our section 2.2) that studied benefits of model merging, a part of our motivation particularly comes from the analysis in Vlaar et al. 2022 which suggested that initializing a model on a higher loss surface—obtained from the height of the barrier in the linear interpolation of models, rather than using random initialization—led to a network achieving better test accuracy. We’ve added this explanation in the paper.
> - We have added plots for visualizing loss landscape and test error (Figure 1 and Figure 6), where we show how unlearning in this particular manner puts parameters on a high loss surface and further training from this point could result in the discovery of more favorable regions in the loss landscape.
> - We have added ablation analysis in Appendix A.3.6 where we use more than one permutations in Interpolate and merge them as suggested by the reviewer. We observed that the number of permutations has minimal impact on overall performance.
> - We have added another ablation in Appendix A.3.5 where we evaluate our method across different interpolate weights with learning rates: (i) 0.1 (ii) 0.01. We observe that while with larger learning rate, varying $w$ has minimal effect on overall performance, with smaller learning rate, $w=0.6$ works best in maintaining plasticity and $w=0.9$ diverges on later tasks.
> - Similar to section 4.1,  we’ve also added a comparison between generalization performance and jump in training loss for increasing total activation score of randomly selected neurons in Appendix A.3.8. We observe that Interpolate results in relatively more efficient adaptation to new tasks, while resetting to random noise results in performance loss when applied to highly active neurons.
> - We have further included a sensitivity analysis in the Appendix A.3.9 to examine the impact of hyperparameter choices on Interpolate and Interpolate+Redo. The results confirm that for the Interpolate method, higher k values generally results in better performance. In contrast, for Interpolate+Redo, no consistent trend emerges, as different combinations perform well. But, we also  observed that an exceptionally higher dormancy threshold for the Redo component negatively affects performance.
>
>
> Tiffany J Vlaar and Jonathan Frankle. What can linear interpolation of neural network loss landscapes
> tell us? In International Conference on Machine Learning, 2022.
>
> ----
>
>
> **Reviewer’s Comment: “this work could be more strengthened by adding experiments in the reinforcement learning setting.”**
>
> **Response:** Thank you for the suggestion. While we agree that extending the work to reinforcement learning (RL) would be an interesting direction for future research, it is currently outside the scope of this study. Our motivation primarily comes from model merging techniques used in supervised learning and fine-tuning, with a primary focus on improving generalization in these contexts. RL scenarios introduce additional complexities and we hope to explore these in future.
>
>
> ----

---

> ### Author Response · Authors · 2024-11-23
> **Response 2/2**
>
> **Reviewer’s Comment: “Section 4.1: The correlation seems weak, and even my statement could not be guaranteed with only one seed. ”**
>
> **Response:** The plots indicate results for different numbers of neurons across five seeds i.e., for a given number of neurons selected for reset, we’ve used five different to select those neurons randomly and trained them to obtain the performance. While the correlation might seem weak, the idea is to suggest that resetting more active neurons based on model merging can interestingly even improve generalizability of the model in contrast to methods like Redo (also see Appendix A.3.8).
>
> ----
>
>
> **Reviewer’s Comment: “can the effect that the authors present also be seen on more larger/complex model sizes/architectures?”**
>
> **Response:** Thank you for your suggestions. We’ve added ResNet exp with CIFAR100 in Appendix A.3.1. We observe that either Interpolate or Interpolate+ReDo, exhibit competitive/better performance suggesting that resetting active neurons can also help maintain plasticity contrary to earlier assumptions.
>
> ----
>
>
> **Reviewer’s Comment: “Can the authors clarify when is the point the model undergoes reset for the experiments in Section 4.2?”**
>
> **Response:** The reset period is fixed at 5 epochs on each experiment. We’ve clarified it in the revised manuscript.
>
> ----
>
>
> **Reviewer’s Comment: “May be the authors could try different reset functions.”**
>
> **Response:** Thank you for the suggestion. We’ve added an experiment in Appendix A.3.7, involving re-initialization (instead of re-randomization) and adding noise instead of using Interpolate. We observed that both variants exhibit relatively worse performance after the hyper-parameter search. We have also added shrink and perturb in Figure 4 which involves shrinking and adding noise to all model parameters. We observe that it only performs better for CNN on Shuffled CIFAR10 and Permuted CIFAR10.
>
> ----
>
>
> **We greatly appreciate your review We have incorporated other minor corrections suggested by the reviewer. We hope that the above responses address the reviewer’s concerns. If the reviewer feels the same, it would be appreciated if this could be reflected through an updated score. If the reviewer has any remaining questions, please do not hesitate to ask, and we will do our best to answer them as soon as possible.**

---

> ### Comment · Reviewer_siNw · 2024-11-25
>
> Thank you for your comprehensive response.
> I see several insights based on the authors rebuttal, such as qualitatively verifying the advantage of Interpolate method by observing the loss landscape (weakness 1). Also, the authors lifted some of my concerns; verification of the ablation on model permutation (weakness 1), results on more larger scale (question 2), and other minor points (weakness 4, question 1, 3). However, I couldn't find a clear explanation resolving weakness 2 and 3, which I believe are the critical points to be addressed in this work.
>
> Also, according to the updated Appendix A.3.7, it is quite non-trivial that trying different reset functions (with keeping the same active neurons to reset) and then Interpolate does not always brings (at least competitive) better performance, seeing from the curve of Re-init (active) in Figure 12. From the nuance from the title and the abstract, it conveys as 'resetting' active neurons is a major factor to this effect, which seems contrary. My intention of this question 4 was comparing different reset functions given in the additional references, while keeping the same active neurons, to figure out that 'resetting' active neurons itself is a major factor or Interpolation plays a more higher role.
>
> Nonetheless, authors resolved several minor concerns, and if the clarification of the above points are addressed, I will be happy to increase my score. Also, please correct me if I have overlooked part of the authors' response regarding weakness 2, 3 and question 4.

---

> ### Author Response · Authors · 2024-11-27
> **Authors' Response**
>
> We’d like to thank the reviewer for replying to our rebuttal and acknowledging the insights from our new experiments. They’ve greatly helped us in improving the paper.
> We’re also glad to know that the reviewer found our responses convincing and that our rebuttal had resolved some of their raised concerns. We’d address your additional concerns below:
>
> **Weakness 2:**
>
> **Response:** We would like to direct the reviewer’s attention to the results in Section 4.2, which we believe validate our hypothesis regarding “the new gradients with higher magnitude…” as discussed in Section 3.2 (previously lines 197–199). As shown in the gradient norm plot, the gradients (averaged across all parameters) obtained by resetting the active model using Interpolate (dark blue) exhibit a higher magnitude compared to Redo (dark red), which primarily resets dormant neurons. We have added this clarification to the paper for greater clarity.
>
>
> ------
>
>
> **Weakness 3:**
>
> **Response:**
> We would like to emphasize that the new visualizations of the loss landscape were included to further substantiate our claim that Interpolate indeed places parameters on a higher loss surface. Training further from this point helps discover regions with better generalization. In both Figure 1 and Figure 6, we observe that Model A (before the jump, green cross) and Model B (Permuted) are located in functionally equivalent loss regions. The resulting “reset” (or jump) using Interpolate (Model C, black cross) positions the parameters on a surface that is at a higher loss region (darker red). After training from this reset point, the Interpolated model reaches a better loss region (blue square in the darker blue region), generally outperforming the model that did not undergo a reset (red square in the lighter blue region).
> While we believe that the 2D contour plots and our analysis in section 4.2 were sufficient to effectively support our hypothesis, we’d like to ask for the reviewer’s opinion if we should still add the plots for more metrics such as Grad norm, weight norm, dormancy per layer, etc. before/after the reset.
>
>
> ------
>
>
> **Question 4:**
>
> **Response:**
> We understand your concern. Based on our experiments in Section 4.1 and the addition of new experiments, we can indeed conclude that interpolate plays a more significant role compared to resetting active neurons. As suggested by other reviewers, we have decided to update the title and abstract to reflect this finding. Our proposed title and abstract are as follows:
>
> *New title*: Interpolate: How resetting neurons with model interpolation can improve Generalizability in Online Learning
>
> *Modified lines in the abstract*: We introduce a model merging approach called Interpolate and show that contrary to previous findings, resetting even the most active parameters using our approach can also lead to better generalization. We further show that Interpolate can perform similarly or better compared to traditional resetting methods, offering a new perspective on training dynamics in non-stationary settings.
>
>
> ------
>
>
>
> We hope that our responses and additional analysis have addressed your concerns. If so, we’d greatly appreciate it if you would increase your score.

---

> > ### Comment · Reviewer_siNw · 2024-11-28
> >
> > I highly appreciate the authors extensive efforts including additional experiments and updates on the manuscript.
> > I have also went through the reviews from the other reviewers, and I found that Reviewer DoQJ shares similar concerns with me (point related to question 4). While the authors have updated the title and the abstract, I also agree that the main experimental settings are misaligned, and the main part of this manuscript should be rewritten. Below are my suggestions to resolve the remaining concerns after reading the authors additional responses.
> >
> > 1. Thank you for clarifying the author's responses related to weakness 2. I understand that the results of Figure 3 shows significant difference in grad_norm between Interpolate (active: 20%) and Redo (inactive: 20%), wouldn't it be more appropriate to report this measure as separate findings? As this result is aimed to support one of the main hypothesis that authors are trying to make. I would say, to make this result in align with the updated title and abstract, it should involve comparing the grad_norm (1) between random (active) and random (inactive), (2) between (active) and (inactive) for various methods, (3) across different methods (especially for Interpolate) for active case, It would've been more convincing. Although, while I still have concerns that only observing the grad_norm measure is not sufficient to support the hypothesis (previously lines 197–199).
> >
> > 2. I appreciate the detailed explanation Figure 1, 6 regarding weakness 3. What I'm certain is that both Figure 3 and Figure (1, 6) are used to support the same message; how appropriate is Interpolate as a point to reset. While Figure (1, 6) focuses on qualitative analysis on across interpolated and the permuted model, and the current Figure 3. focuses on quantitative way between Interpolate and other methods. While I cannot directly give an answer to the authors since the whole manuscript needs rewriting, and its answer would depend on the direction of rewriting, I think it would be more clear if the comparing subject is aligned for the metrics (namely Figure 3), and the loss landscape (namely Figure 1, 6). Additionally, minor suggestion, if the authors will discuss both the effect of utility function and reset function in their updated manuscript, I think separating these would significantly increase the clarity for the readers.
> >
> > 3. I think the main message regarding question 4 is conveyed overall.
> >
> > In summary, I recognize the potential importance of the work itself, as well as the extensive work done by the authors during the discussion phase. Accordingly, I would raise the score to 5, but I do not think the current version is yet adequate to meet the acceptance bar.

---

> ### Author Response · Authors · 2024-11-28
>
> We’d like to thank the reviewer for replying to our response and deciding to increasing the score (although it doesn't appear to be reflected as it still says 3). We’re glad that our responses and rebuttal have addressed the reviewer’s concerns and the reviewer recognizes the potential importance of our work. We address your additional concerns below:
>
>
> **Weakness 2 - Response:**
> - Thank you for your suggestion. Unfortunately, updating the paper to report grad_norm analysis as separate findings would not be possible before today's deadline. While we believe that these findings are sufficient to support our hypothesis which indeed mentions about expecting a higher order gradient magnitude with Interpolate, we’d like to assure the reviewers that we will report this measure as separate findings and also include additional analysis suggested by the reviewer in the Appendix of camera-ready version in order to convey the message even more convincingly.
>
> ------
>
> **Weakness 3 - Response:**
> - We’d like to highlight that, based on the concerns raised by the reviewer (and Reviewer DoQJ), we have revised the manuscript—particularly the title, abstract, and introduction—to clearly convey the key message of our paper. This message, supported by the findings in Figures 1, 3, and 6, *challenges the prevailing narrative in the plasticity research community that predominantly focuses on dormant neurons*. We sincerely appreciate the suggestions, and we also believe that incorporating the additional experiments has already strengthened both our work and the clarity of its presentation.
> - We also appreciate the reviewer’s suggestion on separating the effect of reset function and utility function. However, we’d like to argue that this separation was already done in the presented analysis of section 4.1 which focused only on the reset function and section 4.2 which focused on utility functions.
>
> -----
>
> We hope that our responses have addressed your additional concerns. If so, we’d greatly appreciate it if you would increase your score further.

---

> > ### Comment · Reviewer_siNw · 2024-11-28
> >
> > First of all, thank you for the prompt response. I would like to address the authors response as follows.
> >
> > Authors' response to weakness 2
> > - While I partly agree with the authors that measuring grad_norm across different settings in Figure 3 could support the authors claims (as well as result of right subplot of figure 2), I still do not think it provides strong evidence. How about tracking the norms (or the grad_norms) for active/inactive neurons separately for a toy model (or the changes of the representations itself before and after reset as per proposed in my initial review, note that these are my naive suggestions). While I'm not strongly certain about this, maybe trying these kinds of analysis could be helpful to address the 'beneficial mechanism' that Reviewer ygv8 has made.
> >
> > Authors' response to weakness 3
> > - I think the message regarding weakness 3 in my previous response was not fully conveyed to the authors. I acknowledge that the section 4.1 aims to focus on reset function and section 4.2 focuses on utility functions. What I meant was that the analysis in section 4.1 and section 4.2 is rather weak to stand alone as a section to provide a comprehensive analysis to reset and utility functions correspondingly. It needs reordering in the previous/updated results, and also major rewriting of the manuscript overall. For example, comparing grad_norms between reset functions would further support in parallel with the results of Figure 2 (right) as a different metric to measure. Also, to make an in-depth analysis with respect to the reset function, the ablations regarding the different reset functions should have been reported in section 4.1 (measuring the metric that the authors proposed; jump in training loss, grad_norm).
> >
> > Overall, I still believe that the insights from this work have a certain contribution to the community. However, I'm maintaining my score for now, as there is still room for improvement. If there are any points that I have misunderstood, please let me know.

---

### Official Review · Reviewer_DoQJ · 2024-11-03

**Soundness:** 2
**Presentation:** 2
**Contribution:** 2
**Rating:** 5
**Confidence:** 4

**Summary:**

This work proposes a new plasticity injection method called “Interpolate” for neural networks. It periodically chooses top-k active neurons and readjusts the corresponding model parameters based on “model merging”: it takes an average between itself and a functionally equivalent parameter (established by exploiting permutation invariance). The authors evaluate their method over three types of distribution shifts on CIFAR10 datasets, using MLP with three hidden layers and CNN with two convolutional layers.

**Strengths:**

S1. This work contains an interesting empirical finding. It reveals that the most active neurons can be the subjects of resetting, like the most dormant neurons (e.g., ReDo), for injecting adaptability to a neural network. This is intriguing because, as far as I know, a common belief on why resetting works is that it “awakes” the dormant neurons. On the contrary, this work shows that readjusting the most active neurons can also be beneficial for maintaining plasticity.
S2. As far as I know, it is new to apply model merging (based on the permutation invariance of neural networks) when partially resetting the model to enhance the plasticity.

**Weaknesses:**

Overall, there seems to be a huge room for improvement in the writing and presentation of the paper, so I lean to reject it. Below, I list the reasons.

**W1. The paper lacks intuitive motivation or a mechanistic understanding of the proposed method.**

- To claim that the methodology is useful and powerful, an academic paper should explain **why** it works by providing the following: motivation, underlying mechanism, extensive ablation study on every component of the method, and theoretical guarantees (if possible). However, I cannot find any intuitive motivation or rigorous verifications of why the proposed method works. As I understood, the paper only proposes a method and exhibits its numerical performance.
- Resetting the dormant neurons, as mentioned in the introduction, may mitigate the loss of plasticity because it “recycles dormant neurons back into an active state to recover some of the network’s capacity.” Then, why would unlearning the most active neurons help models maintain their plasticity? Is there any underlying idea about it?
- In Section 4.1, the authors demonstrate that taking an average (i.e., interpolation) of two functionally equivalent model parameters might be a better resetting method than random resetting in terms of the test-time performance. They provide a plausible reason for this: random noise results in greater forgetting. However, if that is the case, is the interpolation a unique answer? I don’t think so. Then why did the author choose to take interpolations? (Let me extend the discussion about the implementation choice later in **W2**) In short, the authors should have provided the reason for the particular implementation choice of their resetting method to convince the readers.

**W2. The experimental results in the paper are insufficient to explain the proposed method’s effectiveness.**

- As admitted by the authors, their resetting method is quite unnatural and too specific. Several variants of the proposed resetting method exist, which should have been tested and compared with their original method.
    - The number of functionally equivalent parameters to take an average: why two? It can be three or more.
    - Mixing ratio: why half and half? It can be generalized to $\lambda \theta_{\tt perm} + (1-\lambda) \theta$, or even generalized to $\sum_{i=1}^m \lambda_i \theta_i$ where $\sum_{i=1}^m \lambda_i = 1$, $\lambda_i > 0$ for all $i$, and $\theta_i$’s are all functionally equivalent parameters to the original $\theta$.
    - The choice of $\theta_{\tt perm}$: why is it uniformly randomly chosen? Even though there are several functionally equivalent parameters, they are only identical regarding the function value. They might not be equivalent in terms of the loss landscape (e.g., gradient, sharpness). Note that there is an optimization technique called teleportation [1], which also exploits the parametric invariance to accelerate the optimization or enhance the generalization capability. Based on this idea, can we choose a “better” $\theta_{\tt perm}$ to maintain the network’s adaptability and generalizability?
    - If the random resetting harms the networks’ generalizability due to a sudden change in active parameters, which induces greater forgetting, what if we add a small noise to the original parameter? Or should we blend the model parameter only with functionally equivalent parameters instead of random noises?
    - More comparisons with other resetting methods, such as Shrink & Perturb [2], resetting the last few layers [3,4], or DASH [5], would strengthen the authors’ contributions.
- Figure 2: it is somewhat questionable that the authors did not perform any hyperparameter tuning here when they compared several variants of Interpolate and ReDo. The default hyperparameter might be unexpectedly/fortunately tailored to “Interpolate (active: 20%)”. For a better comparison, I think the hyperparameter search must be done here. Also, I don’t see why the paper chose particular values for resetting ratios (5% or 20%). It would be nice to provide more ablations on resetting ratios, or the paper should explain why they present only the 5% and 20% results.
- Figure 3: it should display the results of 100%/0% and 0%/100% as well for better visualization. Also, what if the combining ratios do not sum to 100%, i.e., what if there are non-reset parameters?
- Statistical significance: the paper only reports averages over five random seeds. To showcase the stability and statistically significant guarantees of the proposed methods, it would be nice to report medians, interquartile means (IQMs), and confidence intervals (See [6]).
- More experimental details are necessary. For example, the paper should report the batch size, the exact model architecture (e.g., whether normalization layers are used), and running time (i.e., wall-clock time).
- The page limit was up to ten pages for the main text. However, this paper consumes only eight pages for the main text, two for references, and two for the appendix. Although this is not a direct reason for the rejection, I can’t help but suspect the insufficiency of the paper’s content in passing the acceptance bar.

**W3. Minor comments on typos and writing issues.**

- The term “model merging” is a bit misleading. It often refers to a method of ensembling several models to maintain or improve the overall performance of the model(s) [7]. On the contrary, this paper uses the term for a resetting/unlearning strategy.
- Misleading notation in Algorithm 1:
    - $D$ is originally an input dataset but turns into a list of dormancy scores.
    - “for $t=1,2,\ldots,|H|$”: I guess $i$ must be used instead of $t$.
- Until Section 3, the letter $k$ refers to the *number* of neurons to be reset. However, in Section 4.2, the $k$ becomes the *proportion* of the neurons to be reset.
- The caption of Figure 2: “Apply Interpolate” $\rightarrow$ “Applying Interpolate”.
- Appendix A: the last sentence has double full stops (”..”).

---

**References:**

[1] Zhao, Bo, et al. "Improving Convergence and Generalization Using Parameter Symmetries." ICLR 2024.

[2] Ash, Jordan, and Ryan P. Adams. "On warm-starting neural network training." NeurIPS 2020.

[3] Zhou, Hattie, et al. “Fortuitous forgetting in connectionist networks.” ICLR 2022.

[4] Nikishin, Evgenii, et al. “The primacy bias in deep reinforcement learning.” ICML 2022.

[5] Shin, Baekrok, et al. “DASH: Warm-Starting Neural Network Training in Stationary Settings without Loss of Plasticity.” arXiv:2410.23495 (2024).

[6] Agarwal, Rishabh, et al. “Deep reinforcement learning at the edge of the statistical precipice.” NeurIPS 2021.

[7] Yang, Enneng, et al. "Model Merging in LLMS, MLLMs, and Beyond: Methods, Theories, Applications and Opportunities." arXiv:2408.07666 (2024).

**Questions:**

**Q1. Decision of hyperparameters**

- The main hyperparameter of Interpolate is $k$, the number of neurons to apply the interpolation resetting. Practically, how should we determine the value of $k$?

**Q2. Why can resetting the most active neurons achieve comparable performances as resetting the least active neurons?**

- Can we achieve better generalizability with resetting neurons (than without resetting) regardless of their dormancy?
- Have you tried the random selection of neurons to reset? It could be absolutely random or structurally random (e.g., randomly select a layer and reset them all).
- On top of that, why does the vanilla SGD (i.e., not resetting any parameters at all and warm-starting the network at the beginning of every task, if I understood correctly) exhibit a similar performance as the resetting-based method? This seems quite different from the existing observations (e.g., [2], bringing the same numbering of the reference as in Weaknesses).

**Q3. Degradation due to the larger number of epochs**

- In Figure 3, if the learning algorithms run for 10 epochs per task, the optimal combination of ReDo (inactive) and Interpolate (active) seems 10%/90%. However, when increasing the number of epochs per task, the performance significantly drops and this combination becomes the worst. Why does it happen?
- Because of this, I am worried that the optimal combination ratio of ReDo (inactive) and Interpolate (active) might be highly sensitive to the training setups.

**Q4. The exact definition of “permutation function”**

- Section 3.2 illustrates how the interpolation-based resetting works, except for explaining the exact form of the permutation function. What does it look like in a math equation form?

---

> ### Author Response · Authors · 2024-11-23
> **Response 1/3**
>
> We’d like to thank the reviewer for the evaluation of our paper. We’re glad that the reviewer found our empirical finding interesting and our model merging method novel. We also appreciate the suggestions and provide our responses below:
>
> **Reviewer’s Comment: “The paper lacks intuitive motivation or a mechanistic understanding of the proposed method. … why would unlearning the most active neurons help models maintain their plasticity? ”**
>
> **Response:** Thank you for your feedback. We would like to address the reviewer’s concerns as follows:
> - We have added plots for visualizing loss landscape and test error (Figure 1 and Figure 6), where we show how resetting parameters with Interpolate on a high loss surface and further training from this point could result in the discovery of more favorable regions in the loss landscape.
>
> - The motivation comes from the analysis in Vlaar et al 2022 which suggested that initializing a model on a higher loss surface—obtained from the height of the barrier in the linear interpolation of models, rather than using random initialization—led to a network achieving better test accuracy. We’ve added this explanation in the paper. We verify this through our analysis in Sections 4.2 and 4.3, which demonstrates that interpolating a greater number of active neurons can yield results that are comparable to, or even better than, those achieved with Redo.
>
>
> ----
>
>
> **Reviewer’s Comment: “the authors should have provided the reason for the particular implementation choice of their resetting method to convince the readers.
> ”**
>
> **Response:** Thanks for pointing this out. We’d like to clarify that interpolation (model merging) is a well-established idea in the deep learning literature. It has shown a tremendous promise in improving overall generalization capabilities of the model as mentioned in section 2.2. While it is not a unique answer, the reason we employ this in particular is to challenge the existing idea of reviving dormant neurons by randomly initializing them. Interpolation offers a way to avoid entirely forgetting the previously learned parameters even if they are dormant.
>
> We also provide further ablations by comparing different types of resetting neurons apart from Interpolate and random initialization. In particular, we’ve added an experiment in Appendix A.3.7 that uses random selection of neurons and resets them. Overall after an exhaustive hyper-parameter search, selecting neurons randomly and resetting using Interpolate still results in high performance.
>
>
> ----
>
>
> **Reviewer’s Comment: “The experimental results in the paper are insufficient to explain the proposed method’s effectiveness. …”**
>
> **Response:**
> We would like to address the reviewer’s concerns as follows:
> - Thank you for the suggestion. We have added ablation analysis in Appendix A.3.6 where we use more than one permutations in Interpolate and merge them. We observed that the number of permutations has minimal impact on overall performance
> - Thank you for the suggestion. We’ve added ablation with other convex combinations A.3.5 where we evaluate our method across different interpolate weights with learning rates: (i) 0.1 (ii) 0.01. We observe that while with larger learning rate, varying $w$ has minimal effect on overall performance, with smaller learning rate, $w=0.6$ works best in maintaining plasticity and $w=0.9$ diverges on later tasks.
> - Given the exponentially large number of possible permutations, uniformly random selection of permutations is a reasonable choice. As noted by Simsek et al. (2021), the local loss landscape remains largely similar under such random permutations. To provide additional insights, we include 2D plots illustrating the characteristics of the local landscape on a permuted loss surface. We also thank the reviewer for highlighting teleportation [1], which proposes finding grouped elements by constructing an objective function that maximizes the gradient without altering the loss. Additionally, several existing works (Ainsworth et al. 2022, Entezari et al. 2021), have explored optimizing specific objective functions, such as maximizing barrier size, to identify an optimal $\theta_{perm}$. While these approaches offer promising directions, we believe they could serve as potential future extensions to our setup. Incorporating such optimization-based techniques introduces additional complexity compared to randomly selecting permutations, which aligns with the goals of our current work.
> - Thank you for the suggestion. We’ve added an experiment in Appendix A.3.7, involving re-initialization (instead of re-randomization) and adding noise instead of using Interpolate. We observed that both variants exhibit worse performance.
> - We have also added shrink and perturb in section 4.4. We observe that it only performs better for CNN on Shuffled CIFAR10 and Permuted CIFAR10.
>
>
> ----

---

> ### Author Response · Authors · 2024-11-23
> **Response 2/3**
>
> **Reviewer’s Comment: “Figure 2: it is somewhat questionable that the authors did not perform any hyperparameter tuning”**
>
> **Response:**
> Please note that the analysis experiments in Section 4.2 are designed to validate our hypothesis within a fixed set of hyperparameters. We selected k values of 5% and 20% in this particular analysis because these are commonly used and have shown competitive/optimal performance in our experiments.
>
> However, we have included a sensitivity analysis in the Appendix A.3.9 to examine the impact of hyperparameter choices. The results suggest that for the Interpolate method, higher k values generally results in better performance. In contrast, for Interpolate+Redo, no consistent trend emerges, as different combinations perform well. But, we also  observed that an exceptionally higher dormancy threshold for the Redo component negatively affects performance.
>
>
> ----
>
>
> **Reviewer’s Comment: “Figure 3: it should display the results of 100%/0% and 0%/100% as well for better visualization….”**
>
> **Response:** Thank you for pointing it out, we have added 100%/0% curves in Figure 3. However, 0%/100% resulted in divergence in all three cases and is therefore omitted from the figure. In our main experiments involving an exhaustive hyper-parameter search, we do not constrain the top/bottom thresholds to sum to 100%. In fact, both thresholds were tuned independently.
>
>
> ----
>
>
> **Reviewer’s Comment: “Statistical significance”**
>
> **Response:**  Thank you for pointing it out. We have incorporated standard deviation in our results.
>
>
> ----
>
> **Reviewer’s Comment: “More experimental details are necessary.”**
>
> **Response:** Thank you for pointing it out. We’ve added more details in the Appendix A.1.
>
>
> ----
>
>
> **Reviewer’s Comment: “The main hyperparameter of Interpolate is $k$, the number of neurons to apply the interpolation resetting.”**
>
> **Response:** This is indeed the main hyperparameter which is equivalent to the dormancy threshold used in Redo. The only difference is instead of putting constraints on activation score, it is used to select the top/bottom percentile of neurons based on their activation score.
>
> ----
>
>
> **Reviewer’s Comment: “Why can resetting the most active neurons achieve comparable performances as resetting the least active neurons?”**
>
> **Response:** In section 4.1, we indeed do not use any criteria to select neurons. Since the goal of the experiment was to directly compare Interpolate and Random initialization based on the number of neurons selected, we pick the neurons randomly and reset them using Interpolate/random initialization.
> However, to verify exactly how resetting random neurons performs, based on the reviewer’s suggestion, we have added this experiment in Appendix A.3.7, where we observed that Interpolate with both active and random neuron selection perform similarly.
>
>
> ----
>
>
> **Reviewer’s Comment: “Why does the vanilla SGD … exhibit a similar performance as the resetting-based method? ”**
>
> **Response:** The reason SGD baseline exhibits similar performance as reset-based methods is that we have used L2 regularization in our optimization setup which itself has been shown to be a competitive method to address plasticity loss. On top of that, we have conducted an exhaustive huge search over all hyper-parameters to obtain the best-performing setup for each baseline.
>
>
> ----
>
>
> **Reviewer’s Comment: “Degradation due to the larger number of epochs:”**
>
> **Response:**
> We would like to address the reviewer’s concerns as follows:
> - The gradual performance increase over 10 epochs suggests underfitting on individual tasks but positive overall forward transfer. Conversely, training for 100 epochs led to overfitting, and the aggressive resetting associated with the Interpolate method resulted in worse performance on subsequent tasks. This behavior, however, is not a significant issue, as the experiment is designed as an ablation study to understand the relative contributions of the two resetting methods (Redo and Interpolate) in improving performance. In practical scenarios, resetting 100% of parameters is rarely applied, as doing so after full convergence would result in forgetting.
> - To further clarify, we have added a brief hyperparameter search experiment (Appendix A.3.4) using the 100-epochs-per-task setting to demonstrate that the Interpolate+Redo method can achieve performance comparable to other methods. Additionally, we have included a sensitivity analysis (Appendix A.3.9) to examine the impact of hyperparameters. The results indicate that for the Interpolate method, higher k values generally lead to better performance. In contrast, no consistent trend emerges for Interpolate+Redo, as different parameter combinations perform well. However, we observed that setting an exceptionally high dormancy threshold for the Redo component negatively impacts performance.
>
>
> ----

---

> ### Author Response · Authors · 2024-11-23
> **Response 3/3**
>
> **Reviewer’s Comment: “Q4. The exact definition of “permutation function””**
>
> **Response:** We borrow the definition of permutation function and barrier function from Simsek et al. 2021. We’ve updated the text accordingly.
>
> ----
>
>
>
> **Reviewer’s Comment: “The term “model merging” is a bit misleading.”**
>
> **Response:** We’d like to argue that the term “model merging” has been used for the class of methods that try to combine models in order to utilize their individual capabilities together to achieve some goal. In a similar manner, we propose a method that combines multiple models located at functionally equivalent low loss surfaces such that only a subset of features are modified while the remaining ones remain untouched. We show that training from such a resulting model (midpoint) can also ultimately result in an improved performance.
> We also believe that using the term “reset” might have caused confusion. To address this, we have added a clarification in the introduction, specifying that by ‘reset,’ we specifically mean re-locating the parameters to a different point on the loss landscape. This definition of resetting includes any form of re-locating, including model merging.”
>
>
> ----
>
> Berfin Simsek, François Ged, Arthur Jacot, Francesco Spadaro, Clément Hongler, Wulfram Gerstner, and Johanni Brea. "Geometry of the loss landscape in overparameterized neural networks: Symmetries and invariances." In International Conference on Machine Learning, pp. 9722–9732. PMLR, 2021.
>
> Ainsworth, Samuel K., Jonathan Hayase, and Siddhartha Srinivasa. "Git re-basin: Merging models modulo permutation symmetries." arXiv preprint arXiv:2209.04836 (2022).
>
> Rahim Entezari, Hanie Sedghi, Olga Saukh, and Behnam Neyshabur. "The role of permutation invariance in linear mode connectivity of neural networks." arXiv preprint arXiv:2110.06296, 2021.
>
>
> ----
>
>
> **We greatly appreciate this review, whose feedback will help us improve our work. We have also incorporated other minor corrections suggested by the reviewer. We hope that the above responses address the reviewer’s concerns. If the reviewer feels the same, it would be appreciated if this could be reflected through an updated score. If the reviewer has any remaining questions, please do not hesitate to ask, and we will do our best to answer them as soon as possible.**

---

> ### Comment · Reviewer_DoQJ · 2024-11-25
> **Thank you + Remaining questions (1)**
>
> Thank you very much for providing a detailed response! Sorry for the late subsequent reply. It was exciting to have a lot of discussion points in this research direction, as provided by the authors and the other reviewers. Also, thank you for providing extensive ablation studies to address the reviewers’ concerns including mine.
>
> However, there are still remaining questions about the author’s rebuttal and the revised manuscript. Here I list them, mostly following the order of discussion points provided by the author’s rebuttal.
>
> ### **Why does unlearning the most active neurons help?**
>
> - I have some questions/comments on Figure 1:
>     - Is the model the unique permuted variant of model A? What if B is located somewhere else resulting in bad test accuracy even after resetting the model at C?
>     - I don’t understand why the model kept training from A does not minimize the “train” loss. At first glance, I thought the loss landscape was not for the same task where the model is trained. However, Appendix A shows the optimization trajectory and the loss landscape for both “Task 2.” Although SGD cannot fully guarantee the minimization of test error, shouldn’t it decrease the training loss well?
>     - Minor comments on unnecessary capitalizations: Train loss and Test error.
> - Questions on Figure 6:
>     - I’m not really convinced that Interpolation always helps to discover a more generalizable region. For example, the seed 1 result shows that the red trajectory seems to eventually reach a better area than the end of the blue trajectory.
>     - Can you provide the actual numerical values for test errors, by using small texts inside the contour plots?
>     - Together with Figure 1: these plots “may” verify that “Interpolation is a good method” (while I’m not really convinced yet). But can these plots explain “WHY Interpolation is good”? I’ve asked “why” but these figures do not really answer my question in my opinion. If they were the trials to answer “why”, I feel like this explanation is something like ”they can do better generalization because they generalize better.”
> - I guess the authors provide a citation “Vlaar et al 2022” in order to explain the underlying intuition behind their Interpolation method, which is unfortunately unclear. Is it “What Can Linear Interpolation of Neural Network Loss Landscapes Tell Us?” by Vlaar and Frankle (2022)?
> - I don’t really understand why the findings by Vlaar and Frankle (2022) can explain the idea of this work. Could you provide exactly where in Vlaar and Frankle (2022) explaining that “initialization at a point with higher loss leads to better test accuracy”? As I briefly inspected the cited paper, it doesn’t mention the following two points:
>     - Permutation-equivariance of neural network (why do we choose a functionally equivalent configuration of the model?)
>     - Neuron activity (why do we choose the most active neurons to reset? — it is in fact my key question)
>
> ### **Reasons for the implementation choice**
>
> - Did you mean by “Interpolation offers a way to avoid entirely forgetting the previously learned parameters even if they are **active**”? Then I think it makes sense. Still, this cannot fully capture the authors’ finding that adding small noises, which also prevents entirely forgetting the previous knowledge up to some extent, performs significantly worse.
>     - By the way, I’m also curious that the “adding noise” setup (in Figure 12) added the noise only to the most active neurons.
> - I find it interesting that “selecting neurons randomly and resetting using Interpolate still results in high performance.” However, I’m concerned that they somehow disproved(?) their own claim that “resetting the most active neurons might be helpful” through their own experiments…? Still, this result shows that resetting the dormant neurons is not the only way to improve plasticity.
> - The method “Redo with random neuron selection” can be called a variant of “Redo” even though it is no longer a method regarding “dormant” neurons..? Isn’t it just a resetting random proportion of the neural net? Please correct me if I’m wrong.
>
> (continued to the next comment)

---

> ### Comment · Reviewer_DoQJ · 2024-11-25
> **Thank you + Remaining questions (2)**
>
> ### **(Previous) Insufficiency in experiments**
>
> - First of all, thank you again for providing exhaustive additional experiments.
> - About interpolation weight $w$ (Appendix A.3.5):
>     - I am quite surprised that $w$ has an almost negligible effect on test accuracy. Does the benefit of Interpolate occur at $w>0$ sharp? This doesn’t seem natural.
>     - The caption in Figure 11 seems wrong: it says “with smaller learning rate, $w= 0.5$ works best…” but in the rebuttal it says 0.6 is the best. I guess 0.6 is correct, while the differences from the other values are minor.
> - About S&P result:
>     - With my bear eyes, the S&P’s performance seems remarkable for CNNs + no label noise settings. Then someone might say: “If there’s no severe amount of label noise, why don’t we just choose CNN + S&P?”.
>     - Minor typo in Figure 5: it says “Inteprolate” rather than “Interpolate” in the legend. Also, there are duplicates of “Inteprolate+Redo.”
>
> ### **(Previously) No hyperparameter tuning (Figure 3 in the current version)**
>
> - My original concern was exactly about testing only on a fixed set of hyperparameters. But I guess in this case the authors are using the word “hyper-parameter” to represent the main parameters of their method like the resetting ratio $k$ (i.e., the hyperparameters in Table 4), instead of the other hyperparameters like learning rate and batch size (in Table 3)…? Doesn’t the expression “without any hyper-parameter search” mean that, for example, using the fixed values of learning rate and batch size? If so (and I guess so), I meant this might be problematic to compare the methods properly: some of these fixed hyper-parameters might be tailored to certain methods. However, now I don’t think this is a huge problem because of the hyper-parameter search done in Section 4.4.
>
> ### **(Previous) lack of 100%/0% & 0%/100% results (Figure 4 in the current version)**
>
> - Why would the 0%/100% result in divergence? Isn’t it a simple resetting of the whole network at the beginning of every task?
> - Although you don’t constrain the top/bottom thresholds to sum to 100% and tune both thresholds independently, Figure 4 only displays the results where both thresholds are sum to 100%, doesn’t it? I mean, for example, where are the other configurations like 30%/30%? If Figure 14 in Appendix A.3.9 is partially showing such results, then the main text (or at least the caption of Figure 4) should have a pointer to it.
>
> ### **Last Comments**
>
> - Overall, I am still not fully convinced by the motivation of the proposed method. It seems to me that there is no explicit reason why resetting the most active neurons might be beneficial for preventing the loss of plasticity. The experimental improvements, which are not very impressive, seem to be discovered by chance and provide no clear messages on what is really important for retaining the model’s continual adaptability. The main reason is that the author’s additional experiments (on randomly choosing neurons to reset) reveal that in fact resetting the most active neurons is not a key to enhancing plasticity (more or less contradicting the paper’s title).
> - However, the idea of applying a convex combination of functionally equivalent parameters as a resetting method is quite intriguing. The paper could be better than the current submission if the paper focused more on the topic “model interpolation as a resetting method” by discussing deeper, e.g., “which parameters should we take interpolation with?” or “why are the permutation variants the good choice to take interpolation with?”.
> - In this sense, I still lean to ‘reject’ rather than ‘accept’. If the authors further address my questions here, I might raise my score to 5 but not higher.

---

> ### Author Response · Authors · 2024-11-28
> **Authors' Response 1/4**
>
> We’d like to thank the reviewer for replying to our rebuttal and acknowledging a new set of extensive ablation studies that have further strengthened our paper. We address your concerns below:
>
> **Figure 1**
> **Response:**
> We’d like to address the reviewer’s questions/concerns regarding Fig 1 as follows:
> - Yes, Model B is a unique permuted variant of Model A. Since both Model A and Model B are functionally equivalent, they should lie in similar/equivalent regions on both the training and test loss surfaces. However, the interpolated Model C can lie in any type of region, which, as our empirical results show, generally corresponds to a higher loss surface.
> - We would like to clarify that the model trained from A using SGD does indeed minimize the training loss, as indicated by its path reaching a lighter blue region, which represents a lower loss surface compared to the higher loss surface (dark red region). However, since Model C (using SGD + Interpolate) ultimately reaches an even lower loss value, the contour plot assigns a relatively darker blue tone to the region reached by SGD + Interpolate. While SGD decreases the training loss, it essentially gets stuck in a suboptimal region that is not as low or optimal as the one reached by SGD + Interpolate when trained until convergence.
>
> -------
>
>
>
> **Figure 6**  **Response:**
> We’d like to address the reviewer’s questions/concerns regarding Fig 6 as follows:
>  - We agree with the reviewer that Figure 6 illustrates the tendency of SGD in the online learning setup to get stuck in suboptimal regions, even when a more generalizable solution exists. The reason we plotted test surfaces for all five seeds was to demonstrate this phenomenon. However, when combined with Interpolate, we empirically found that the chances of reaching a more generalizable and optimal region increased. Our extensive analysis further supports this claim, where Interpolate (or Interpolate + Redo) outperforms the SGD baselines, as shown in experiments on Shuffled CIFAR-10 with both MLP and CNN architectures. These results further motivate the need for a deeper theoretical investigation into Interpolate and other reset-based methods in non-stationary settings.
> - Regarding the purpose of Figures 1 and 6, these are intended to illustrate how Interpolate resets the parameters, moving them to a higher loss surface, from which further training can lead to the discovery of more generalizable regions. These figures are not meant to directly address “why Interpolate is good.” We attempt to answer this question in Section 3.2 through our unlearning hypothesis and the idea of “new gradients with higher magnitude.” We empirically validate this in Section 4.2. As shown in the gradient norm plot, the gradients (averaged across all parameters) obtained by resetting the active model using Interpolate (dark blue) have a higher magnitude compared to Redo (dark red), which primarily resets dormant neurons. We have added this clarification to the paper.
>
> -------

---

> ### Author Response · Authors · 2024-11-28
> **Authors' Response 2/4**
>
> **Regarding Vlaar et al 2022 -** **Response:**
> We’d like to address the reviewer’s questions/concerns regarding our reference to Vlaar et al 2022 as follows:
> - We refer to Figure 3 in “What Can Linear Interpolation of Neural Network Loss Landscapes Tell Us?” by Vlaar and Frankle (2022) where it is observed that initialization at a point with higher loss improves the test accuracy.
> - We would like to direct the reviewer’s attention to line 243 in our updated manuscript to clarify that we do not claim the findings of Vlaar et al. (2022) fully explain our work. Instead, we draw parallels to their observation that initializing at a higher loss point can lead to better test accuracy. At the same time, we agree with the reviewer that Vlaar et al. (2022) did not address permutation-equivariance or neuron activity. However, to address the reviewer’s concern, we’ve added the following in the Introduction which clarifies the motivation further:
>    - *Permutation-equivariance*: Our motivation comes from an extensive literature on linear mode connectivity and loss barrier analysis which suggests a link between low-loss barriers between minima with training stability and generalization [1]. Several approaches have been proposed to improve linear mode connectivity by reducing loss barriers between minima in order to improve generalization through model merging techniques [2,3]. However, [4] showed that such loss barriers between minima can be minimized cost-effectively by exploiting the permutation invariance property of neural networks. By resetting the model on high-barrier regions, we propose our method Interpolate which utilizes permutation invariance to reset parameters in non-stationary settings which intentionally introduces controlled instability, acting as a regularizer. We hypothesize that training from this reset point would allow SGD to navigate toward a more stable loss region, ultimately improving generalization.
>    - *Neuron activity*: The rationale for leveraging higher neuron activity is based on the hypothesis that resetting neurons with higher activity will lead to a more significant increase in training loss. This hypothesis is empirically validated through our analysis, which is detailed in Appendix A.3.8.
>
> ----
>
> **Implementation choice -** **Response:**
> - Yes, the reviewer is correct in understanding our statement “Interpolation offers a way to avoid entirely forgetting …. “.
>    - We would also like to clarify in response to the reviewer’s comment regarding adding small noise to the original parameters. We conducted an ablation experiment, detailed in Appendix A.3.7, where noise was added to the entire model (rather than just the active parameters). This baseline resulted in worse performance, and we have now included this setup detail in the paper. While adding an experiment that introduces noise only to the active neurons may not be feasible before the deadline, we would appreciate the reviewer’s opinion on whether such an experiment should be conducted for further discussion.
> - We respectfully disagree with the reviewer’s assessment of our claim that “resetting the most active neurons might be helpful.” Prior to our investigation, research in online learning was largely dominated by the belief that only dormant neurons should be perturbed or reset to improve plasticity. While our initial motivation was centered around resetting active neurons, our analysis not only demonstrated that this approach is effective but also validated it by introducing Interpolate. We agree, however, that our additional analysis opens up the possibility for further investigation into this type of model merging technique, which can show competitive performance without relying on activation scores.
> - Yes, “Redo with random neuron selection” is in fact just a resetting random proportion of the neural net and can be called a variant of Redo. We’ve added this as a baseline upon the suggestions from other reviewers.
>
>  -----

---

> ### Author Response · Authors · 2024-11-28
> **Authors' Response 3/4**
>
> **Interpolation weight -** **Response:**
> - We would like to clarify that the results in Figure 11 were obtained through an exhaustive hyper-parameter search over all parameters, except for the learning rate and w . However, we agree that it is indeed surprising that w has almost a negligible effect. To investigate this further, we’ve provided additional analysis for different learning rates, which shows a clear trend at smaller learning rates. Specifically, for w = 0.9  (which moves parameters closer to the permuted model), we observe that the model diverges on later tasks, while the optimal values for  w  are around 0.5 or 0.6—these values even outperform  w = 0.1! Additionally, we would like to note that the ablation experiment was only conducted on Shuffled CIFAR-10. While we won’t be able to add similar experiments with other benchmarks before the deadline, we’d be happy to conduct them if the reviewer believes they are important for continuing the discussion. We would appreciate your opinion on whether such experiments with other benchmarks would significantly improve the paper.
> - Thank you for pointing it out. We’ve corrected the Figure 11 caption.
>
> ------
>
>
> **S&P result -**  **Response:**
> - Yes, S&P is indeed remarkable in the CNN + no label noise settings. However, we would like to argue that S&P is not a direct baseline for our work, as we have specifically limited the scope of our paper to “reset-based methods that rely on neuron activity.” In contrast, S&P modifies the entire network without considering neuron activity, making it a different approach. We believe that S&P could act as a complementary method to ours. Combining S&P with Interpolate, or even applying S&P specifically to active neurons, would be an interesting direction for future work.
> - Thank you for pointing it out. We’ve corrected it in the updated paper.
>
>
> -----
>
> **Figure 3 -**  **Response:**
>  - By “without any hyper-parameter search”, we indeed meant using the fixed values of learning rate and batch size. But, as we mentioned in our earlier response, this was meant to be an analysis experiment to directly compare the methods to investigate their behavior deeper. We’ve highlighted this point further in the updated paper.
>
> -----
>
>
> **Figure 4-**  **Response:**
> - These analysis experiments were conducted under a limited compute budget of 10 and 100 epochs per task. Resetting the entire model (0%/100%) using Redo every 5 epochs (reset period) did indeed lead to divergence. We’d like to emphasize that we used SGD with momentum, with a fixed learning rate and momentum state. Since the parameters were being reset too frequently, the training from momentum states carried over from previous epochs could have contributed to this divergence. To overcome such issues in our actual benchmarking experiment in Section 4.4, we included the reset period and momentum coefficient as hyper-parameters in the exhaustive search.
> - Thank you for your suggestion. We’ve added a pointer to the appendix.
>
> -----
>
>
> **Reviewer’s Comment: "The experimental improvements, which are not very impressive”**
>
> **Response:**
> We’d argue that it is well-established in online learning research that no new techniques clearly outperform all other baselines in preventing the loss of plasticity without significantly modifying the model (e.g., through changes to activation functions or layer normalization) (Lyle et al., 2024). In fact, Lyle et al. (2024) suggested that L2 regularization with layer normalization is generally sufficient to perform similarly or even better than other methods. Therefore, without modifying the model, we incorporated L2 regularization by default in all our experiments, which resulted in a highly competitive SGD baseline.
>
> We’d also like to highlight that, unlike our experimental setup, several recent works proposing plasticity methods have shown improvements without conducting an exhaustive hyper-parameter search such as ours and have typically evaluated performance based solely on online training accuracy, rather than test accuracy, as we do in our work.
>
> -------

---

> ### Author Response · Authors · 2024-11-28
> **Authors' Response 4/4**
>
> **Reviewer’s Comment: The paper could be better than the current submission if the paper focused more on the topic “model interpolation as a resetting method”...”**
>
> **Response:**
> We understand the reviewer’s concern. Based on our current experiments, newly added experiments, and suggestions from other reviewers, we propose to update the title and abstract accordingly as follows:
>
> *New title*: Interpolate: How resetting neurons with model interpolation can improve Generalizability in Online Learning
>
> *Modified lines in the abstract*: We introduce a model merging approach called Interpolate and show that contrary to previous findings, resetting even the most active parameters using our approach can also lead to better generalization. We further show that Interpolate can perform similarly or better compared to traditional resetting methods, offering a new perspective on training dynamics in non-stationary settings.
>
> In addition to that, we’ve also updated our introduction to be more focused on the above topic and added motivation for our method as suggested by the reviewer.
>
> -------
>
> [1] Jonathan Frankle and Michael Carbin. The lottery ticket hypothesis: Finding sparse, trainable neural networks. arXiv preprint arXiv:1803.03635, 2018.
>
> [2] Seyed Iman Mirzadeh, Mehrdad Farajtabar, Dilan Gorur, Razvan Pascanu, and Hassan Ghasemzadeh. Linear mode connectivity in multitask and continual learning. arXiv preprint arXiv:2010.04495, 2020.
>
> [3] Norman Tatro, Pin-Yu Chen, Payel Das, Igor Melnyk, Prasanna Sattigeri, and Rongjie Lai. Optimizing mode connectivity via neuron alignment. Advances in Neural Information Processing Systems, 33: 15300–15311, 2020.
>
> [4] Rahim Entezari, Hanie Sedghi, Olga Saukh, and Behnam Neyshabur. The role of permutation invariance in linear mode connectivity of neural networks. arXiv preprint arXiv:2110.06296, 2021.
>
>
>
>
>
> -------
>
> We hope that our responses and additional updates in the paper have addressed your concerns. If so, we’d greatly appreciate it if you would increase your score.

---

> > ### Comment · Reviewer_DoQJ · 2024-11-28
> > **Raising my score to 5 + Remaining questions**
> >
> > Thank you for your detailed responses resolving some of my concerns. I'm glad to see that the paper has been improved significantly. However, plenty of the main concerns remain the same.
> >
> > ### **Figure 1**
> >
> > - What I meant by “minimizing the train loss” is literally finding the (local) minima of the train loss or an even darker blue region. Why would the SGD trajectory be stuck on a light blue region, i.e., a suboptimal region in terms of training loss? Of course, SGD can get stuck to stationarity, but the light blue area doesn’t look like a stationary region. Does this mean that SGD is NOT fully trained until convergence as SGD+Interpolate? If so, is it fair to compare these methods with test errors? If a model is not learned well, then how can we expect it to generalize well? Is it because the model size is too small?
> > - The caption should mention that the green and purple hidden nodes are selected to be permuted; hence the distinct permutation variant B of A is unique.
> >
> > ### **Vlaar & Frankle (2022)**
> >
> > - First of all, it is weird to cite their work as ‘Vlaar et al.’, which seems there were three or more authors. This is why I got confused with the citation.
> > - I would say, the high-loss initialization itself implies nothing about a test-time performance, until we exactly characterize the shape of the loss landscape. They might have some correlation, but as you know, the correlation is not causation. I guess the “linear interpolation” provides a meaningful structure, but I’m not sure.
> > - I think the author’s argument that “we choose the most active neurons when we reset the model with interpolation because resetting active neurons leads to high-loss re-location of the model parameter, inducing better test-time performance” is wrong. I think the argument should proceed with the other way around, something like “if we want to reset the active neurons and while preventing the loss of plasticity, we should apply model interpolation.” In other words, the authors should’ve told “we should apply interpolation when we reset the most active neurons” rather than saying “we should reset the most active neurons when we use interpolation” (which is the author’s original argument, I guess).
> >
> > ### **S&P**
> >
> > - In my opinion, it doesn’t make sense to argue that “S&P is not our scope because it doesn’t utilize neuron activity.” This is because the neuron activity is useful side information that we can use; even if we have one, we can choose whether we use it or not. So it is natural to say that “by taking advantage of side information, we can make the model better than ones not using this information.” That is, it is a bit awkward to say “since we utilize more information, the methods using less information are not in our scope even though they are significantly better than us.”
> >
> > ### **Change in title and abstract + Final remarks**
> >
> > - Thank you for incorporating the suggestions raised by myself and other reviewers. The new title and abstract seem much better.
> > - Nonetheless, the main argument of the paper, the main methods of the paper, and the main experiments of the paper still focus a lot on resetting the most active neurons. I think it is not a significant contribution because resetting the same amount of randomly chosen neurons can achieve similar performance, showcased via the author’s experiments. In my opinion, the whole message of the paper should be like “Interpolation is a good resetting method for maintaining the neural network’s plasticity, even when we reset the most active neurons.” The title, abstract, and introduction are adequately modified accordingly, but I think the whole experiment section must be rewritten and reorganized to support the message.
> > - Considering the significant improvement in the paper during the discussion period and the necessity of potential rewriting, I raise my score to 5, which is still below the acceptance bar.

---

> ### Author Response · Authors · 2024-11-28
>
> We’d like to thank the reviewer for replying to our response and raising the score. We address your concerns below:
>
> **Figure 1 -** **Response:**
>
> We’d like to address reviewer’s remaining questions/concerns regarding Fig 1 as following:
> - We’d like to clarify that the plotted loss surface is an approximation which was computed by taking an average over 40 batches (40*128/40k training samples) for the loss function and computed on 100 * 100 models. Since testing on a full dataset was not possible, the reason light blue area doesn’t look like a stationary region is that the averaged contour plots interpolate between sharp minima if they exist in the surface. However, we still believe that it is fair to compare these methods with test errors because they’re evaluated when the model starts overfitting because model size is small. We've added these details in the paper.
> - Thank you for the suggestion, we’ve updated the caption accordingly.
>
>
> ------
>
> **Vlaar and Frankle (2022) -** **Response:**
> - We’d like to counter-argue that based on the experiments conducted in the paper, our statement about *choosing the most active neurons when we reset the model with interpolation* acknowledges the finding that resetting active neurons is sufficient to achieve the best performance with Interpolate. On the other hand, stating “we should apply interpolation when we reset the most active neurons” implies that interpolation is, in fact, *necessary* when resetting the active neurons - which we believe is not the case. This is because achieving competitive/better performance with Interpolate itself challenges the prevailing narrative that tends to focus predominantly on dormant neurons in the plasticity research community — which is the key contribution of our paper.
>
>
> ------
>
>
> **S&P-** **Response:**
> - We’d again like to highlight that while S&P performs better on CNN with Shuffled CIFAR10 and Permuted CIFAR10, these are still 1/3rd of all main experiments. It still is one of the worst/second-worst methods on MLP+Shuffled CIFAR10, MLP+Shuffled CIFAR10 and CNN+Noisy CIFAR10. On the other hand, Interpolate (or Interpolate + Redo) is either the best / equally best / second best performing approach in all our experiments.
> - We believe these results and our arguments align with the core message of our paper, which we'd like to re-iterate -challenging the prevailing narrative in the plasticity research community that predominantly focus dormant neurons.
>
>
> -----
>
> **Reviewer’s Comment: “resetting the same amount of randomly chosen neurons can achieve similar performance.”**
>
> **Response:**
> - We respectfully disagree with the reviewer’s assessment with the new results  “resetting the same amount of randomly chosen neurons can achieve similar performance”. While we believe it is indeed an interesting result, it still does not undermine the main message of our paper - “resetting dormant neurons is not necessary for improving plasticity and even resetting active neurons can achieve such results”. We’ve shown this by proposing Interpolate which we believe opens up the possibility of exploring such methods further in the future.
> - We’d also like to highlight that the ablation was conducted only on Shuffled CIFAR10 with MLP settings. While Interpolate with active neurons performs similar to Interpolate with random neurons, it still performs marginally better which aligns with our observation in A.3.8 where our analysis suggested that resetting neurons with higher activity with Interpolate empirically performs (although marginally) better.
> - Since we believe that the ablation experiment conducted only on MLP + Shuffled CIFAR10 is not enough evidence to claim that “resetting the same amount of randomly chosen neurons can achieve similar performance” and since the reviewer is still concerned about these findings, we’re conducting additional experiments with other benchmarks and models to investigate this to a larger extent.
>
> *<NEW UPDATE>*: Upon conducting the same experiment on Permuted CIFAR10 (Figure 14 in the updated manuscript), we find that the best performing setup for Interpolate (on random neurons) performs slightly worse than Interpolate (on active neurons). We will run additional baselines, report/refine our analysis similar to Figure 12 for the camera-ready version.
>
>
> ------
>
> We hope that our responses have addressed your most of your questions/concerns. If so, we’d greatly appreciate it if you would increase your score further.

---

> > ### Comment · Reviewer_DoQJ · 2024-11-28
> >
> > Thank you for leaving the response promptly.
> >
> > * I don’t think challenging an existing idea can solely be the contribution of an academic paper. However, unfortunately, the authors are arguing that “dormant neurons are not the only subject to reset” is their KEY contribution.
> >   * Honestly, I'm not sure whether it's true that the plasticity research community is "predominantly focused on" dormant neuron resetting, as the authors keep claiming. This is because the only two prior works I already knew about resetting methods based on neuron activity are ReDo and CBP, which are the only two prior works the authors brought in their paper. Are two papers enough to be regarded as the "prevailing narrative in the research community"? I guess not, considering the massive and fast development in this area.
> >
> > * Moreover, although the authors’ last response and additional experiments answer some of my questions, my original thought about the necessity of rewriting didn’t change. Although the paper is significantly improved compared to the initial version (which is why I raised the score to 5), I believe it has much room for improvement as the other reviewers mostly agree (which is why I do not increase the score anymore). Therefore, I still would like to keep my score. However, I look forward to seeing a much-improved version of this work at another future conference, which may include the enhanced originality of their methods, a more intuitive and plausible explanation of the underlying mechanism of their methods, and extensive experimental results they promised to put in their manuscript.

---

### Official Review · Reviewer_ygv8 · 2024-11-06

**Soundness:** 2
**Presentation:** 2
**Contribution:** 2
**Rating:** 5
**Confidence:** 4

**Summary:**

This paper introduces a resetting-based method to address the plasticity loss issue in non-stationary learning settings, such as online learning. The authors empirically demonstrate that resetting the most active parameters can yield performance improvements, contrasting with previous findings. The method, called Interpolate, combines this reset-based approach and model merging. Empirical experiments on the CIFAR-10 dataset show that this method achieves competitive performance when compared to other approaches.

**Strengths:**

- This paper provides a clear background, making it accessible to readers unfamiliar with the problem.
- The approach of resetting active/inactive parameters is intriguing.
- The authors compare the proposed method with other approaches such as ReDo and CBP across different types of distribution shifts on a benchmark dataset (CIFAR-10).

**Weaknesses:**

- The authors need to conduct additional experiments to thoroughly test their hypothesis and validate the proposed method. The current results are limited to a few experiments on the CIFAR-10 dataset with relatively simple neural network architectures, which raises concerns about the robustness of these findings. Testing on a broader range of datasets (e.g., CIFAR-100 or noisy datasets), more complex network architectures (e.g., ResNet or other high-capacity networks), alternative utility functions, and different problem domains (e.g., deep reinforcement learning) would provide stronger empirical support for the claims made in the paper

**Questions:**

### Major
- While the improvements via resetting the most active parameters are intriguing, the paper does not explain why these effects occur, particularly in contrast with the original ReDo paper. I am curious whether the dormant neuron phenomenon observed in ReDo is also present here. Additionally, while this paper uses the dormancy score as a utility function, I am uncertain whether the authors’ claims hold when alternative metrics (e.g., Zhou et al. [1]) are used to assess neuron importance

[1] Zhou, Hattie, et al. "Fortuitous forgetting in connectionist networks." arXiv preprint arXiv:2202.00155 (2022).

- The ratio between $\theta$ and $\theta_{perm}$ in Eq. (2) likely has a substantial impact on neural network learning dynamics. The authors used a midpoint merge between $\theta$ and $\theta_{perm}$, but it would be informative to understand how different ratios might affect outcomes

- I am unclear why the authors did not use the optimal hyper-parameters from Sec. 4.4 in the experiments shown in Fig. 2. Using consistent hyper-parameters could improve result interpretability.

- The phenomenological findings on the impact of resetting active parameters appear loosely connected to the proposed Interpolate method. While this relationship might align with the question raised in lines 50-51, it is known that ReDo is not the only way to reactivate a model. A clearer explanation of how these findings connect would strengthen the paper


### Minor
- In line 172, it is unclear why parameters $\phi_k$ are in $\mathbb{R}^k$.
- The experimental details provided are insufficient for full comprehension. While references are given, this paper should contain specific details on the datasets used (lines 221-222), task composition, definitions for terms like ‘reset with random noise’ in Sec. 4.1, ‘Dormancy’ in Fig. 2, etc.
- In Sec. 4.1, how many runs were averaged to obtain the results? Also, the term ‘jump in training loss’ in Fig. 1 is not clearly defined.
- Please provide the hyper-parameters used for each method in Figs. 2 and 3.
- Why does the number of tasks (x-axis) differ in Fig. 3?
- In line 188, there exists a typo: $pi_k$ should be $\pi_k$.

---

> ### Author Response · Authors · 2024-11-23
> **Response 1/2**
>
> We’d like to thank the reviewer for the evaluation of our paper about the intriguing explanation and empirical analysis presented in our method. We also appreciate the suggestions and provide our responses below:
>
> **Reviewer’s Comment: “Testing on a broader range of datasets … architectures”**
>
> **Response:** Thank you for your suggestion.
>
> - We’ve added ResNet exp with ResNet and CIFAR100 in Appendix A.3.1. We observe that either Interpolate or Interpolate+ReDo, exhibit competitive/better performance suggesting that resetting active neurons can also help maintain plasticity contrary to earlier assumptions. We’ve also added experiments on Permute MNIST in Appendix A.3.2.
> - While we agree that extending the work to reinforcement learning (RL) would be an interesting direction for future research, it is currently outside the scope of this study. Our motivation primarily comes from model merging techniques used in supervised learning and fine-tuning, with a primary focus on improving generalization in these contexts. RL scenarios introduce additional complexities and we hope to explore these in future.
>
> ----
>
>
> **Reviewer’s Comment: “whether the dormant neuron phenomenon observed in ReDo is also present here (in Interpolate).”**
>
> **Response:**  Yes, the second plot in section 4.2 shows that dormancy is present and increases (similar to online) in case of Interpolate since it doesn’t perturb the dormant neurons. To validate whether this might hurt the performance on future tasks, we’ve also added results with more tasks in Appendix A.3.2 showing that increase in dormancy does not play a huge role in overall behavior and Interpolate still shows similar performance as other baselines.
>
> ----
>
>
> **Reviewer’s Comment: “I am uncertain whether the authors’ claims hold when alternative metrics”**
>
> **Response:**  We'd like to point the reviewer to Interpolate (CBP) results shown in section 4.2 where we use CBP's utility function (instead of dormancy) and use Interpolate as the reset function.
>
> However, we’ve also added an experiment in Appendix A.3.7 that uses random selection of neurons (similar to Zhou et al. [1]) and resets them. Overall, after an exhaustive hyper-parameter search, selecting neurons randomly and resetting using Interpolate still results in high performance. Since our goal in this paper was to mainly challenge the conventional approaches of resetting inactive neurons, we believe that this direction could also be explored deeper in future work.
>
> ----
>
>
> **Reviewer’s Comment: “it would be informative to understand how different ratios might affect outcomes”**
>
> **Response:** We’ve added an experiment in Appendix A.3.5 where we evaluate our method across different interpolate weights with learning rates: (i) 0.1 (ii) 0.01. We observe that while with larger learning rate, varying $w$ has minimal effect on overall performance, with smaller learning rate, $w=0.6$ works best in maintaining plasticity and $w=0.9$ diverges on later tasks.
>
>
> ----
>
>
> **Reviewer’s Comment: “optimal hyper-parameters from Sec. 4.4 in the experiments shown in Fig. 2”**
>
> **Response:**
> “The analysis experiments in Section 4.2 are designed to validate our hypothesis within a fixed set of hyperparameters. We selected k values of 5% and 20% in this particular analysis because these are commonly used and have shown optimal performance.
> However, Based on the reviewers’ recommendations, we have included a sensitivity analysis in the Appendix A.3.9 to examine the impact of hyperparameter choices. The results confirm that for the Interpolate method, higher k values generally results in better performance. In contrast, for Interpolate+Redo, no consistent trend emerges, as different combinations perform well. But, we also  observed that an exceptionally higher dormancy threshold for the Redo component negatively affects performance.
>
>
> ----

---

> ### Author Response · Authors · 2024-11-23
> **Response 2/2**
>
> **Reviewer’s Comment: “Impact of resetting active parameters appear loosely connected to the proposed Interpolate method”**
>
> **Response:**
> We would like to address reviewer’s concern with following:
> - A part of motivation for resetting active neurons using model merging comes from the analysis in Vlaar et al. 2022 which suggested that initializing a model on a higher loss surface—obtained from the height of the barrier in the linear interpolation of models, rather than using random initialization—led to a network achieving better test accuracy. We’ve added this explanation in the paper. While a higher loss surface can be obtained by several existing methods (as also suggested by Reviewer DoQJ), they generally involve solving an optimization problem that introduces additional complexity compared to randomly selecting permutations. We have added ablation analysis in Appendix A.3.6 where we use more than one permutations in Interpolate and merge them. We observed that the number of permutations has minimal impact on overall performance.
> - Moreover, similar to our analysis in section 4.1, we’ve added a comparison between generalization performance and jump in training loss for increasing total activation score of randomly selected neurons in Appendix A.3.8. The analysis suggests that Interpolate results in relatively more efficient adaptation to new tasks, while random noise can result in performance loss when applied to more active neurons.
> - Lastly, we’ve added a 2D contour visualization of the loss surface (Figure 1 and Figure 6) illustrating how resetting with Interpolate can help in discovery of a better generalizable region.
>
> ----
>
>
> **Reviewer’s Comment: “In line 172, it is unclear why parameters $\phi_k$ are in $\mathbb{R}^k$.”**
>
> **Response:** Thanks for pointing it out, we’ve updated the line accordingly.
>
> ----
>
>
> **Reviewer’s Comment: “The experimental details provided are insufficient for full comprehension.”**
>
> **Response:** Thanks for pointing it out, we’ve added all details in Appendix A.1
>
> ----
>
> **Reviewer’s Comment: “In Sec. 4.1, how many runs were averaged to obtain the results? Also, the term ‘jump in training loss’ in Fig. 1 is not clearly defined.”**
>
> **Response:** As mentioned in Line 232, five runs were averaged to obtain the results. We have added a line explaining ‘jump in training loss’ in the revised paper.
>
> ----
>
> **Reviewer’s Comment: “Please provide the hyper-parameters used for each method in Figs. 2 and 3.”**
>
> **Response:** Thanks for the suggestion. We’ve added all details about the best hyper-parameter setup obtained from the exhaustive search in A.2. Unless specified otherwise, we’ve used the same hyper-parameters for all our experiments.
>
> ----
>
> **Reviewer’s Comment: “Why does the number of tasks (x-axis) differ in Fig. 3?”**
>
> **Response:** The number of tasks differ in case of Noisy CIFAR10 and Shuffled CIFAR10. This is because Noisy CIFAR10, which was proposed by Lee et al. 2024, divides the dataset into 10 chunks and the label noise of decreasing level is applied on each chunk. Therefore, the number of unique tasks are limited to 10. On the other hand, for shuffled CIFAR10, we generate 100 unique tasks where each task is created by remapping the labels of the original dataset to new labels.
>
> Hojoon Lee, Hyeonseo Cho, Hyunseung Kim, Donghu Kim, Dugki Min, Jaegul Choo, and Clare Lyle. Slow and Steady Wins the Race: Maintaining Plasticity with Hare and Tortoise Networks, June 2024b.
>
> ----
>
> **We greatly appreciate this review, whose feedback will help us improve our work. We have also incorporated other minor corrections suggested by the reviewer. We hope that the above responses address the reviewer’s concerns. If the reviewer feels the same, it would be appreciated if this could be reflected through an updated score. If the reviewer has any remaining questions, please do not hesitate to ask, and we will do our best to answer them as soon as possible.**

---

> > ### Comment · Reviewer_ygv8 · 2024-11-27
> >
> > I thank the authors for their thoughtful responses to my questions, and I apologize for the delay in replying.
> >
> > While the additional explanations and experiments have strengthened the paper, I remain uncertain about the beneficial mechanism behind resetting the neurons using their proposed method. For example, it is unclear why the performance of the Interpolate method does not vary with the ratio $\omega$ in Fig. 11.
> >
> > As a result, I find the novelty of this work insufficient for publication at the conference. There is no clear justification for merging the two methods, and the final performance of the proposed approach is not convincing.
> >
> > Furthermore, the main questions raised by the authors in lines 49–52 are not fully addressed. I believe the authors should either shift their focus to align with their findings or conduct additional experiments to resolve these issues.
> >
> > Accordingly, I will raise my score from 3 to 5, while maintaining that this work remains below the acceptance threshold.

---

> ### Author Response · Authors · 2024-11-27
> **Authors' Response 1/2**
>
> We’d like to thank the reviewer for replying to our rebuttal and acknowledging the newly added experiments that have further strengthened our paper. We’re glad that the reviewer raised their score and would like to address the additional concerns below:
>
>
> **Reviewer’s Comment: “unclear why the performance of the Interpolate method does not vary with the ratio w in Fig. 11. ”**
>
> **Response:** We’d like to clarify that the results in Figure 11 were obtained through an exhaustive hyperparameter search across all parameters, except for the learning rate (lr) and  w . While  w  does not play a significant role when lr = 0.1 , we observe a clear trend with smaller learning rates: when  w = 0.9  (moving closer to the permuted model), the model diverges on later tasks, and the optimal values tend to lie around  w = 0.5  or  0.6 (even better than w=0.1).
> Additionally, we would like to note that the ablation experiment was performed solely on the Shuffled CIFAR-10 dataset. While we won’t be able to add similar experiments with other benchmarks in the paper before the deadline, we'd ask for the reviewer’s opinion if you think they are important to continue the discussion and whether they would meaningfully improve the paper.
>
> ----
>
> **Reviewer’s Comment: “ There is no clear justification for merging the two methods ”**
>
> **Response:** We’re not sure what the reviewer meant by “merging the two methods”.
> - If the reviewer is referring to Section 4.3, where we combined Interpolate and Redo, we’d like to clarify that this experiment was designed as an analysis to investigate which individual method plays a more dominant role in achieving better performance in a constrained setup. However, in Section 4.4, we conducted an exhaustive hyperparameter search to thoroughly evaluate the combined approach against all other baselines. Notably, the combination of these methods has demonstrated competitive or even superior performance in that context.
> - Assuming the reviewer is referring to our model merging technique, we have addressed this in detail in our responses to other reviewers and have updated the manuscript accordingly (line 88-99).
>
> Please let us know the one of the above points addresses the reviewer’s concern. If not, we’d like to ask the reviewer to please clarify the concern further.
>
> ----
>
>
> **Reviewer’s Comment: The final performance of the proposed approach is not convincing ”**
> **Response:** We respectfully disagree. We would argue that in online learning research that no single technique consistently outperforms all other baselines in preventing the loss of plasticity without significantly modifying the model (e.g., activation functions or layer normalization) (Lyle et al., 2024). In fact, Lyle et al. (2024) also suggest that L2 regularization, when combined with layer normalization, is generally sufficient to achieve performance comparable to or better than more complex methods. Therefore, we have incorporated L2 regularization by default in all our experiments, resulting in a highly competitive SGD baseline after an exhaustive hyper-parameter search. We would like to highlight that using Interpolate as a resetting method, our overall idea is to challenge the conventional focus on dormant neurons when addressing plasticity. Our results demonstrate that even resetting the most active neurons—when done correctly—can yield comparable or even superior performance to baselines such as Redo and CBP.
>
>
> ----

---

> > ### Author Response · Authors · 2024-11-28
> > **Authors' Response 2/2**
> >
> > **Reviewer’s Comment: the main questions raised by the authors in lines 49–52 are not fully addressed.**
> >
> > **Response:** We respectfully disagree with the reviewer’s assessment regarding lines 49–52. To address the concerns, we list the research questions mentioned in those lines and highlight the corresponding experiments in the paper that answer them:
> >
> > - *Research Question 1:  Can resetting the non-dormant neurons also improve plasticity?*
> > This is the primary research question that we pose in our paper which is answered collectively in our experiments/analysis in section 4.2 and 4.4.
> >    - In Section 4.2, we show that even as dormancy increases, resetting the most active neurons using Interpolate can achieve strong performance, often surpassing baselines like Redo and CBP, which focus on resetting dormant neurons exclusively.
> >    - In Section 4.4, we conducted an exhaustive hyperparameter search experiment, ensuring a fair comparison by allocating an equal compute budget to our method and all other baselines. These experiments revealed that resetting the most active neurons with Interpolate results in comparable or better performance. Interestingly, Redo did not outperform the SGD baseline at all in general, whereas Interpolate on Shuffled CIFAR-10 achieved clear improvements over Redo for both MLP and CNN architectures.
> >
> > - *Research Question 2: How many parameters should be reset?*
> >    - In our analysis in 4.1, we obtain a trend between the number of parameters to reset using both Random init and Interpolate. The results across 5 seeds for increasing the number of neurons indicate that the higher number of neurons when reset using Interpolate would result in better generalization performance on a new task. On the other hand, when reset to random initialization, the performance would eventually decrease with more neurons selected.
> >    - Our analysis in section 4.3 further suggested that Interpolating a relatively larger number of active neurons (indicated by $k$ percentile) helps in terms of generalizability than Redo.
> >    - To concretize this finding further, the hyper-parameter search experiment in section 4.4 reveals that not only Interpolate perform equally well or better than Redo, but the optimal number of neurons (and corresponding parameters) that resulted in the best performance are obtained from the set [0.05, 0.1, 0.2, 0.5] as shown in Table 5. These numbers clearly show that while the number of parameters to reset can vary across setups, an optimal answer can be obtained.
> >
> > - *Research Question 3: is resetting the parameters associated with dormant neurons the only method to reactivate the model?*
> > We address this question in all our experiments when we compare Redo with our proposed reset method Interpolate. Throughout our paper, we’ve argued and shown that when reset using Interpolate, we reactivate the model in order to improve the generalizability of the model in an online learning setting. However, our experiment in Appendix A.3.7 further addresses this question as we add a baseline of Redo with a random selection of neurons. While this baseline did not outperform Interpolate in the beginning tasks, it still is competitive suggesting that even a random selection of neurons can reactivate the model.
> >
> >
> > ----
> >
> > **Reviewer’s Comment: the authors should either shift their focus to align with their findings or conduct additional experiments to resolve these issues.**
> >
> > **Response:** We believe that the experiments in the main paper and the newly added ablation experiments in the appendix after the reviewer's suggestions clearly answer the research questions (as discussed above) and lie within the scope that we set on in our paper. We have also updated the Title, Abstract, and Introduction recently based on the responses from other reviewers on our rebuttal. However, while adding new results is not possible given the deadline, we would greatly appreciate the reviewer’s input on specific experiments that could further strengthen the paper.
> >
> > ---
> >
> >
> > We hope that our responses have addressed your concerns. If so, we’d greatly appreciate it if you would increase your score further.

---

> ### Comment · Reviewer_ygv8 · 2024-11-28
>
> Thank you to the authors for their responses. The revised manuscript is more appropriately focused on their proposed approach, Interpolate. However, several concerns remain:
>
> ### Lack of a clear beneficial mechanism behind the Interpolate method (main concern)
>
> - The Interpolate method combines (i) the permutation invariance property to reduce loss barriers between minima and (ii) the activity-based reset method to reactivate the model. While the authors have shown that, counterintuitively, resetting active neurons with their method can improve the generalization of neural networks, they have not fully explained the underlying mechanism for this phenomenon or the specific role of interpolation. Additionally, it remains unclear whether the Interpolate method effectively reduces loss barriers as intended on their method. While the approach is intriguing, the discussion and justification are insufficient in the manuscript's current form
>
> ### Concerns Related to the Research Questions
>
> - Can resetting the non-dormant neurons also improve plasticity?
> The provided answer appears to focus specifically on the Interpolate method and does not generalize to other resetting techniques. To thoroughly address this question, the authors should explore the results of various resetting methods, not just Interpolate. Moreover, the rationale behind why resetting non-dormant neurons improves plasticity in the proposed method remains unclear.
> - How many parameters should be reset?
> The results presented are again specific to the Interpolate method. While the optimal number of parameters to reset may vary depending on tasks, architectures, and other factors, the purpose or broader implications of this question are not evident.
> - Is resetting the parameters associated with dormant neurons the only method to reactivate the model?
> As mentioned in my initial review, it is well-known that ReDo is not the only method for resetting parameters to reactivate a model, as cited by the authors in the manuscript. Therefore, raising this as a research question seems unnecessary.
>
> While I appreciate the authors’ efforts in revising the manuscript and providing additional clarifications, the remaining issues prevent me from recommending this work for publication. Accordingly, I will maintain my score below the acceptance threshold.

---

> > ### Author Response · Authors · 2024-11-28
> > **Authors' response 1/2**
> >
> > We’d like to thank the reviewer for replying to our rebuttal. We would like to address the additional concerns below:
> >
> >
> > **Reviewer’s Comment: “... they have not fully explained the underlying mechanism for this phenomenon or the specific role of interpolation. ...”**
> >
> > **Response:**
> > - To clarify the rationale behind resetting non-dormant neurons, we wanted to challenge the prevailing narrative that tends to focus predominantly on dormant neurons in the plasticity research community. We’ve added this in our updated manuscript and also included the following motivation behind using Interpolate in the Introduction: *Our motivation comes from an extensive literature on linear mode connectivity and loss barrier analysis which suggests a link between low-loss barriers between minima with training stability and generalization [1]. Several approaches have been proposed to improve linear mode connectivity by reducing loss barriers between minima in order to improve generalization through model merging techniques [2,3]. However, [4] showed that such loss barriers between minima can be minimized cost-effectively by exploiting the permutation invariance property of neural networks. By resetting the model on high-barrier regions, we propose our method Interpolate which utilizes permutation invariance to reset highly active parameters in non-stationary settings which intentionally introduces controlled instability, acting as a regularizer. We hypothesize that training from this reset point would allow SGD to navigate toward a more stable loss region, ultimately improving generalization.*
> > - To further address the reviewer’s concern, we would like to direct the reviewer’s attention to the results in Section 4.2, which we believe validate our hypothesis regarding “the new gradients with higher magnitude…” as discussed in Section 3.2. As shown in the gradient norm plot, the gradients (averaged across all parameters) obtained by resetting the active model using Interpolate (dark blue) exhibit a higher magnitude compared to Redo (dark red), which primarily resets dormant neurons. We have added this clarification to the paper for greater clarity.
> >
> > We've added the above explanation in the manuscript.
> >
> >
> > [1] Jonathan Frankle and Michael Carbin. The lottery ticket hypothesis: Finding sparse, trainable neural networks. arXiv preprint arXiv:1803.03635, 2018.
> >
> > [2] Seyed Iman Mirzadeh, Mehrdad Farajtabar, Dilan Gorur, Razvan Pascanu, and Hassan Ghasemzadeh. Linear mode connectivity in multitask and continual learning. arXiv preprint arXiv:2010.04495, 2020.
> >
> > [3] Norman Tatro, Pin-Yu Chen, Payel Das, Igor Melnyk, Prasanna Sattigeri, and Rongjie Lai. Optimizing mode connectivity via neuron alignment. Advances in Neural Information Processing Systems, 33: 15300–15311, 2020.
> >
> > [4] Rahim Entezari, Hanie Sedghi, Olga Saukh, and Behnam Neyshabur. The role of permutation invariance in linear mode connectivity of neural networks. arXiv preprint arXiv:2110.06296, 2021.
> >
> >
> > ----

---

> ### Author Response · Authors · 2024-11-28
> **Authors' response 2/2**
>
> **Reviewer’s Comments  on question 1**
>
> **Response:**
> The reason we focused on comparing Interpolate with Redo is that the plasticity research community that focuses on resetting neurons have mostly relied on only one type of reset function which is used in ReDO and CBP, the most well-known methods in this class of methods which are often the baselines used in recently proposed methods.
> However, based on the reviewer’s suggestions, we’ve also included another resetting method - re-initialization in Appendix A.3.7 and have shown that it does not perform as good as Interpolate.
>
> ----
>
>
> **Reviewer’s Comment on question 2**
>
> **Response:** We respectfully disagree with the reviewer’s assessment that implications of this question are not evident from our results because of the fact that there’s an optimal (non-zero) number of non-dormant neurons and resetting them can also result in improved plasticity. These results clearly align with our core message discussed earlier. However, we’d still like to ask for the reviewer’s opinion on whether removing this question from our introduction should make things clearer and would align our findings better. If yes, we’d like to assure the reviewer that we’ll modify that part of the Introduction accordingly in the camera-ready version.
>
> ----
>
>
> **Reviewer’s Comments on question 3**
>
> **Response:** We’d again like to re-iterate to our original argument that the plasticity research community has predominantly focused on the existence of dormant neurons and have mostly relied on only one type of reset function which is used in ReDO and CBP, the most popular baselines in this class of methods. Therefore, we raised this question in the first place to challenge the narrative built around for dormant neurons for reactivating the model.
>
> -----
>
> We hope that our responses have addressed your concerns. If so, we’d greatly appreciate it if you would increase your score further.

---

### Author Response · Authors · 2024-11-23

We thank all the reviewers for their insightful comments and suggestions. We are pleased that the reviewers found our work to present a clear background (Reviewer ygv8), offer interesting empirical findings (Reviewer DoQJ), and provide novel insights to the field through comparative analysis and identifying the impact of utility functions on active neuron selection (Reviewer siNw). We also appreciate the recognition of our contribution as a worthwhile, as noted by Reviewer QsVM. Furthermore, we are glad that the reviewers found the paper well-written and accessible, underscoring its clarity and relevance to the field.


We have responded to specific concerns and list the changes we made to the revised manuscript here:

Section 1:
- Clarified the use of the term ‘reset’ in the paper.
- Added a figure 1 that summarizes our overall idea and further motivates the idea of model merging.

Section 3:
- Made some minor changes related to better explaining the methodology and algorithm as suggested by reviewers.
- Added connection with previous works.

Section 4:
- Added standard deviation.
- Made some minor changes in writing as suggested by reviewers.

Appendix:
- A.1: Added more implementation details including discussion on Figure 1.
- A.1: Added Figure 6 showing more contour plot visualization
- A.2: Added best-performing hyper-parameter setup obtained from the exhaustive search.
- A.3.1: Added hyper-parameter sweep experiments for ResNet18 on all three non-stationary settings with CIFAR100.
- A.3.2: Added hyper-parameter sweep experiments involving training for a large number of tasks on Permuted MNIST and Permuted CIFAR10
- A.3.3: Added hyper-parameter sweep experiments with Adam as base optimizer
- A.3.4: Added hyper-parameter sweep experiments with higher compute budget assigned for each task (100 epochs per task)
- A.3.5: Added ablation experiment for comparing performance of different convex combinations of base and permuted model in Interpolate.
- A.3.6: Added ablation experiment for comparing performance with using multiple permutations of the base model in Interpolate.
- A.3.7: Added experiment with baselines: (i) random selection of neurons, (ii) re-initialization, (iii) adding noise.
- A.3.8: Added an extension of section 4.1 analysis experiment with activation score.
- A.3.9: Conducted sensitivity analysis to examine the impact of hyperparameter choices on Interpolate and Interpolate+Redo.


<<<< Post discussion >>>>>
- Title, Abstract: Slightly modified both after reviewers suggestions.
- Introduction: Added a paragraph for motivating model merging methods proposed in our work.
- Section 4.2: Added explanation on how our analysis on gradient norm validates our hypothesis in section 3.2.
- Figure 14: Additional analysis comparing Interpolate with active neurons selection and random neurons selection on Permuted CIFAR10
- Made some minor corrections on newly ablation experiments.


All the changes made in the revised manuscript are highlighted in blue font.

---

### Meta-Review · Area_Chair_Hs7R · 2024-12-18

**Metareview:**

This paper proposes a method named “Interpolate” aimed at reducing the loss of plasticity, which modifies the parameters of the most active neurons by taking a random permutation of active neurons and then taking an interpolation of the original parameter and the permuted parameter. This method challenges the existing belief that resetting dormant neurons is essential for alleviating the loss of plasticity.

The authors and the reviewers had an exceptionally active discussion period, and the authors put extensive efforts in refining the paper, including supplementing the paper with a lot of additional experimental results. I appreciate the efforts from the authors as well as the reviewers.

My overall evaluation of this paper after reading through the paper, reviews, and comments is that the paper unfortunately falls slightly below the acceptance threshold. While the proposed method offers a new insight on the study of plasticity and shows some reasonable performance against other baselines, the common concern by the reviewers—why does it work—remains largely unresolved, in my opinion. Although the authors point to Vlaar & Frankle (2022), I am not fully convinced why we should expect that starting model training from a higher training loss will result in a better test accuracy. Moreover, the effectiveness of the proposed algorithm seems to depend on specific settings, and even in the settings it outperforms other baselines, the performance gaps do not look statistically significant (e.g., Figure 5). Although the proposed idea is novel and interesting, the paper does not demonstrate sufficient theoretical/empirical merits of the algorithm.

Also, it was pointed out by multiple reviewers that the authors should consider re-positioning their contributions relative to the existing architecture. I concur with reviewers that the word “reset” in the literature includes an implicit meaning that the parameter is randomly re-initialized. Hence, positioning the proposed approach as a “reset” method can cause unnecessary confusion.

In any case, I believe that despite the authors’ extensive efforts (which I am grateful of), the paper still needs some revision to meet the bar for acceptance at ICLR. Hence, I recommend rejection at this time.

**Additional Comments On Reviewer Discussion:**

Other than points mentioned in the summary, some additional comments on the discussion period:
- The initial submission was rather brief and did not include thorough ablation studies. The reviewers asked for additional experiments in many different aspects, and the authors diligently carried out these experiments and updated the paper.
- Reviewer siNw suggested that the paper could be strengthened by adding some RL experiments. The authors responded that this is out of scope, which I understand given the limited time to revise the paper. However, I agree with the reviewer on this point and recommend the authors to consider some RL experiments in the next revision of the paper.

---

### Decision · Program_Chairs · 2025-01-22

Reject